# Automatic Prompt Augmentation and Selection with Chain-of-Thought from Labeled Data

**Kashun Shum**$^{\heartsuit *}$, **Shizhe Diao**$^{\heartsuit *}$, **Tong Zhang**$^{\heartsuit}$
$^{\heartsuit}$The Hong Kong University of Science and Technology
{ksshumab, sdiaoaa, tongzhang}@ust.hk

## Abstract

Chain-of-thought (CoT) advances the reasoning abilities of large language models (LLMs) and achieves superior performance in complex reasoning tasks. However, most CoT studies rely on carefully designed human-annotated rational chains to prompt LLMs, posing challenges for real-world applications where labeled data is available without rational chains. This paper proposes a new strategy, **Automate-CoT** (**Autom**atic Prompt **A**ugmen**t**ation and **Se**lection with **C**hain-**o**f-**T**hought), that can bypass human engineering of CoT by automatically augmenting rational chains from a small labeled dataset, and then pruning low-quality chains to construct a candidate pool of machine-generated rationale chains based on the labels. Finally, it selects the optimal combination of several rationale chains from the pool for CoT prompting by employing a variance-reduced policy gradient strategy to estimate the significance of each example. Automate-CoT enables a quick adaptation of the CoT technique to different tasks. Experimental results demonstrate the effectiveness of our method, where competitive results are achieved on arithmetic reasoning (+2.7%), commonsense reasoning (+3.4%), symbolic reasoning (+3.2%), and non-reasoning tasks (+2.5%).[1]

## 1 Introduction

The recent success in large language models (LLMs) has shown that properly prompted LLMs demonstrate emergent capabilities on complex understanding and question-answering tasks (Wei et al., 2022a). Especially, with the recently proposed chain-of-thought (CoT) prompting (Wei et al., 2022b), LLMs are capable of solving reasoning tasks including arithmetic reasoning, commonsense reasoning, and symbolic reasoning. The basic idea of CoT prompting is adding a few rationale chains to the answer as exemplars to illustrate the intermediate reasoning steps. Following CoT, several recent studies improve it by leveraging self-consistency (Wang et al., 2023), explanation learning (Lampinen et al., 2022), complexity-based prompting (Fu et al., 2023), self-training (Huang et al., 2022), voting verifier (Li et al., 2022a), and bootstrapping (Zelikman et al., 2022).

However, most of them are constrained to a few *fixed* human-written exemplars, which require significant human efforts to create and adapt to new datasets. The annotation process is nontrivial because humans need to not only select the questions but also carefully design the reasoning steps for each question. In the process of searching for the perfect exemplars, we identify four critical factors that affect the performance of chain-of-thought prompting and require large human effort to deal with: (1) order sensitivity (Zhao et al., 2021): the order combination of the exemplars; (2) complexity (Sugawara et al., 2018; Lai et al., 2021; Fu et al., 2023): the number of reasoning steps of the rationale chains; (3) diversity: the combination of different complex-level exemplars; (4) style sensitivity (Papadopoulos et al., 2010): the writing/linguistic style of the rationale chains. Detailed analysis of the four factors is covered in Section 2. All of these sensitivities make human-based prompt engineering costly and motivate us to find an automatic and task-agnostic way to adapt chain-of-thought exemplars to any downstream tasks.

In this paper, we solve these problems by a CoT augmentation and selection process to find suitable exemplars automatically. This can be divided into three steps: (1) Augment: The language model generates multiple pseudo-chains for query questions automatically. (2) Prune: Based on an assumption: *Generating correct reasoning is a necessary condition for generating correct answers.* This assumption is natural because the answer is

---

*Equal Contribution.

[1]The code is available at https://github.com/SHUMKASHUN/Automate-CoT.

generated after several reasoning steps. When a correct answer is generated, the rationale chain of these steps is most likely correct, contributing to the final correctness. As a result, We prune the pseudo-chains according to the consistency between generated and ground-truth answers to reduce the noise. (3) Select: Given that all the data have been annotated with rationale paths, we propose to apply a variance-reduced policy gradient strategy (Williams, 1992; Dong et al., 2020; Zhou et al., 2021; Diao et al., 2022) to estimate the gradients and optimize the selection process to find the most helpful chain-of-thought for each task. Compared to prior manually written CoT, Automate-CoT could find the optimal and diverse CoT automatically, adaptable to any task without human effort. Compared with Auto-CoT (Zhang et al., 2023), which samples diverse questions by clustering and generates rationale chains, Automate-CoT considers and mitigates the aforementioned sensitivity issues, while achieving a greater performance boost for each task. Automate-CoT is a fully automatic pipeline for finding better chain-of-thought prompts, mitigating the sensitivity issues of manually written exemplars, and further improving the performance by a large margin. Experimental results demonstrate the effectiveness of Automate-CoT on arithmetic reasoning (+2.7%), commonsense reasoning (+3.4%), symbolic reasoning (+3.2%), and non-reasoning tasks (+2.5%).

## 2 Motivation

Recent studies observed sensitivity issues of GPT-3's few-shot learning caused by different selections of in-context examples such as order instability (Zhao et al., 2021; Zhang et al., 2022; Liu et al., 2022; Lu et al., 2022). Based on their findings, we first investigate whether these sensitivities still exist in chain-of-thought methods. Then we further explore other factors that would not only affect the performance but require human efforts to deal with. We conclude with the following four factors:

• Order Sensitivity: Different orders of few-shot exemplars may cause a huge impact on the performance in traditional few-shot prompting (Lu et al., 2022). Thus we conduct experiments on GPT-3 to test if there is such sensitivity in chain-of-thought methods. Although Manual-CoT (Wei et al., 2022b) reports that the human-written CoT is robust to order changes (<2%) with the LaMDA model, we observed that the performance of GPT-3

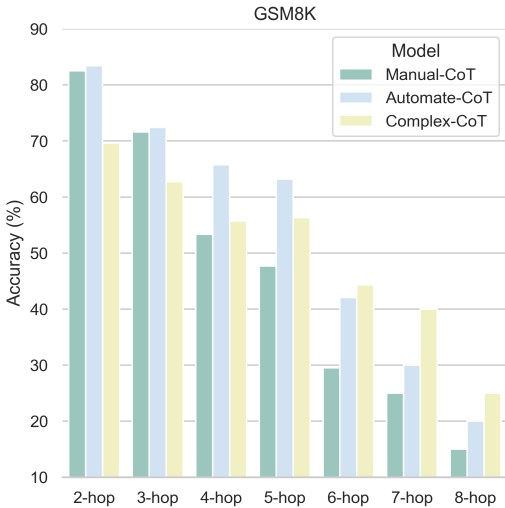

Figure 1: The performance across different numbers of hops (reasoning steps of rationale chains) on GSM8K. Manual-CoT refers to the human-written chain-of-thought by Wei et al. (2022b). Complex-CoT refers to the chain-of-thought using 9-hop rationale chains.

fluctuates with different orders of chain-of-though exemplars. For the GSM8K dataset, we simply randomly shuffle the order of the exemplars in Manual-CoT 10 times and the lowest accuracy can be 59.8% which is 3.3% lower than the average accuracy (63.1%) they report, suggesting that order sensitivity still exists.

• Complexity: We first define complexity as the number of hops (reasoning steps) in an exemplar where more steps indicate greater complexity. It is observed that human-written CoT tends to be simple ($\leq$3 hops), achieving good accuracy in simple math questions while suffering from complex questions, as shown in Figure 1. In addition, a previous study (Fu et al., 2023) suggested that using all complex exemplars can improve CoT performance. However, in our experiments (Figure 1), we found that Complex-CoT can improve the accuracy of complex questions, but perform poorly in simple questions. Therefore, we conjecture that the inconsistency between the hops of provided exemplars and the required hops of the real question causes the performance drop, suggesting that determining the appropriate complexity level of exemplars is crucial.

• Diversity: Based on the above discovery about complexity, a natural question is what combination of different complex-level exemplars is most effective. However, testing various combinations is a challenging task for humans and requires significant effort to determine the optimal one. In our

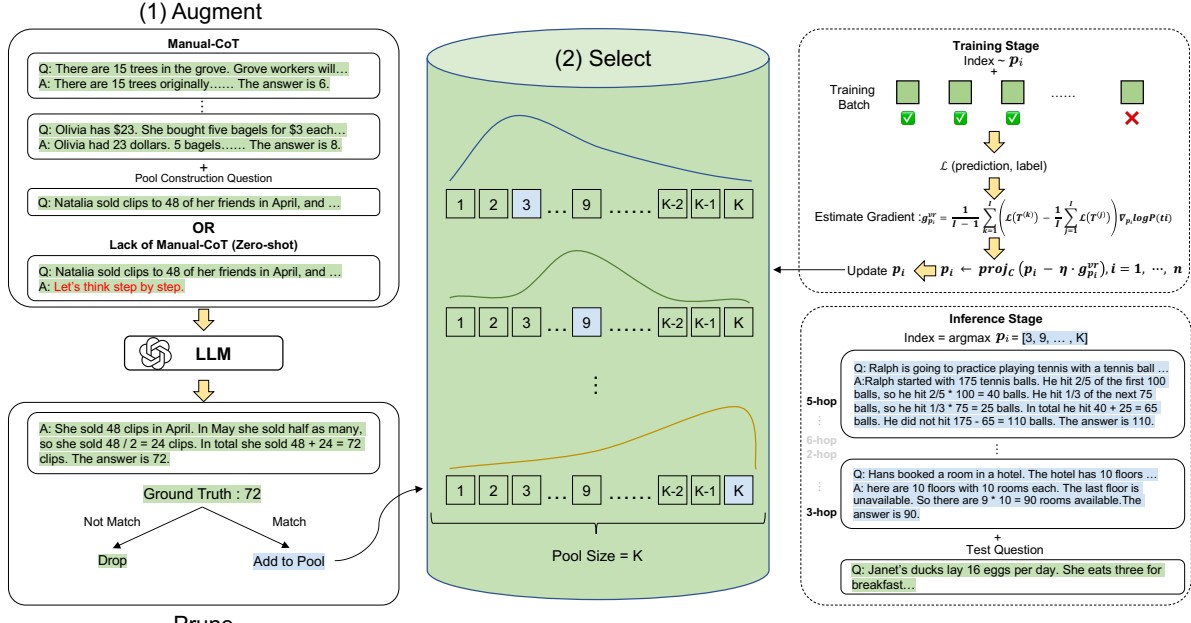

Figure 2: Illustrations of our proposed approach. The left and middle parts of the figure contain two steps of our method: **(1) Augment and Prune** and **(2) Select**. The right part illustrates the training stage (right top) and the inference stage (right bottom), respectively.

experiments (Figure 1), we found that a combination of different complex-level exemplars outperforms CoT with all complex exemplars, suggesting a complexity-diversity trade-off.

• Style Sensitivity: Previous research in educational psychology found that different learning styles would limit the cognitive benefit for students from the prompting (Papadopoulos et al., 2010). We further argue that students with specific learning styles benefit to varying degrees from different styles of prompting. In addition, the empirical evidence from Manual-CoT (Wei et al., 2022b) shows that different annotators can cause up to 28.2% accuracy difference in a symbolic reasoning task, verifying our conjecture. As a result, some *bad* styles may lead to a huge performance drop. However, humans cannot determine the performance of a particular style beforehand, so it requires trial and error by checking on the validation set, which further increases the effort of writing chain-of-thought exemplars.

In light of this empirical evidence, we are motivated to design a framework not only to augment rationale chains but also to select helpful chains adaptively. With this framework, it is expected to bypass the order and style sensitivities and reach a better complexity-diversity trade-off without human effort, finally boosting performance.

## 3 Approach

Our approach receives a training dataset $D$ containing $n$ questions $Q = \{q_1, q_2, ..., q_n\}$, and $n$ answers $A = \{a_1, a_2, ..., a_n\}$. The overall architecture of our approach is shown in Figure 2. In this section, we start with a detailed description of *augment and prune* operation and end with an illustration of *select* operation.

### 3.1 Augment and Prune

Inspired by Wang et al. (2022), which shows that the generated rationale chains are of comparable quality to the human-annotated ones, we aim to automatically generate the rationale chains to augment the candidate exemplars. Given $m$ fixed rationale chains $C = \{c_1, c_2, ..., c_m\}$, a question $q$, we ask the large language model $\mathcal{G}$ to generate $k$ rationale chains for each $q$. A larger k can form a larger pool and some post-processes can be done to improve the quality of the pool. Considering the cost and efficiency, we choose $k = 1$ for our experiments. Our method works well even without $C$ (i.e., $m = 0$), which is based on zero-shot prompting. Then we prune those incorrect ones out and only keep those with the correct final answer. In other words, the final answer should be consistent with the ground-truth answer. After pruning, we

obtain a pool of $K$ high-quality exemplars.

## 3.2 Select

With a large pool of high-quality exemplars, we cannot directly apply all of them due to four considerations: (1) context length limit: the maximum length is 2,048 for GPT-3, so we cannot feed too many exemplars into the model. (2) fair comparison: existing studies usually take 4-8 question-answer pairs as exemplars following Wei et al. (2022b). (3) sensitivity: the model performance may be sensitive to the contexts (Jiang et al., 2020), orders (Lu et al., 2022) and lengths (Lester et al., 2021) from the observation of prompt learning literature. (4) adaptation: different downstream tasks may require different exemplars. Therefore, a natural idea is to select the most suitable 4-8 exemplars automatically.

The process can be deemed as optimizing a supervised model with latent variables. For each chain-of-thought index $i$, we initialize a latent variable $j_i \sim \text{Cat}(\boldsymbol{p}_i)$. The random variable $j_i$ is sampled with the probability distribution $\boldsymbol{p}_i = [p_{i,1}, \cdots, p_{i,N}]$ over the $N$ candidate demonstration indexes, where $\boldsymbol{p}_i \in \mathcal{C}$ and $\mathcal{C} = \{\boldsymbol{p} : \|\boldsymbol{p}\|_1 = 1, 0 \preceq \boldsymbol{p} \preceq 1\}$. Since $\boldsymbol{p}_i$ is independent of each other, the joint probability of the whole input exemplars is $P(T) = \Pi_{i=1}^n P(t_i) = \Pi_{i=1}^n p_{i,j_i}$. The loss is formulated as $\mathcal{L}(\mathcal{G}([T, S], y))$, where $T$ represents the full few-shot exemplars, $t_i$ denotes the i-th exemplar, and $S$ is the current question (user's query). However, directly updating the prompts by back-propagating through $\nabla_{\boldsymbol{p}_i} \mathcal{L}(\mathcal{G}([T, S], y))$ is not possible because of the inaccessible gradients, where $y$ is the label. We resort to the variance-reduced policy gradient estimator (VR-PGE) (Williams, 1992; Dong et al., 2020; Zhou et al., 2021; Diao et al., 2022), a kind of reinforcement learning method to optimize the loss function via forward propagation with:

$$\mathbb{E}_T [\mathcal{L}(T)] = \int \mathcal{L}(T) P(T) \, \mathrm{d}T, \qquad (1)$$

and estimate the gradient of $\boldsymbol{p}_i$ by:

$$\boldsymbol{g}_{\boldsymbol{p}_i}^{vr} = \frac{1}{I-1} \sum_{k=1}^I \left( \mathcal{L}(T^{(k)}) - \frac{1}{I} \sum_{j=1}^I \mathcal{L}(T^{(j)}) \right) \nabla_{\boldsymbol{p}_i} \log P(t_i)$$

(2)

where $T^{(k)}, k = 1, \cdots, I$ are sampled independently from $P(T)$. Therefore, the exemplar distribution $\boldsymbol{p}_i$ can be updated by a projected stochastic gradient descent algorithm:

$$\boldsymbol{p}_i \leftarrow \text{proj}_\mathcal{C}(\boldsymbol{p}_i - \eta \cdot \boldsymbol{g}_{\boldsymbol{p}_i}^{vr}), i = 1, \cdots, n \qquad (3)$$

where $\eta$ is the learning rate, $I$ is the sample size, and $\text{proj}_\mathcal{C}$ is the projection calculation (details are presented in the Appendix A).

## 4 Experimental Settings

In this section, we first introduce the setting of eleven datasets and their corresponding evaluation metrics (§ 4.1). Then the baseline models (§ 4.2) and implementation details (§ 4.3) are presented in the following two subsections, respectively. Full details about the experimental setting are illustrated in Appendix B.

### 4.1 Datasets and Evaluation Metrics

Following Wei et al. (2022b), we conduct our experiments on eight reasoning tasks, including five math word problem datasets: GSM8K, ASDiv, SVAMP, AQuA, and SingleOp; two commonsense reasoning datasets: CommonsenseQA (CSQA) and StrategyQA, and one symbolic reasoning task: Last Letter Concatenation (Letter (4)). We also generalize our method to non-reasoning tasks including one question-answering task (OpenBookQA), one natural language inference task (e-SNLI), and one sentiment analysis task (SST-2). The detailed statistics of the datasets are listed in Table 5. The evaluation metric for all tasks is the exact match accuracy. First, we conduct pre-processing for predictions to remove all the special symbols. For example, "\$100,000" will be processed to "100000". Then we check if it has the same value as the ground truth to calculate the exact match accuracy.

### 4.2 Baselines

We compare our method with the following baseline methods: chain-of-thought (Manual-CoT) (Wei et al., 2022b), self-consistency (SC) (Wang et al., 2023), and Auto-CoT (Zhang et al., 2023). And we utilize the public APIs from OpenAI's services[2] and test with text-davinci-002 and code-davinci-002.

### 4.3 Implementation

**Augment and Prune:** Following Wei et al. (2022b) and Wang et al. (2022), we keep the same number of exemplars (4-8) listed in Table 5. For main experiments, we augment and prune a pool of 100 high-quality exemplars for all datasets.

[2] https://openai.com/api/

| Method | GSM8K | ASDiv | SVAMP | AQuA | SingleOp | CSQA | STQA | Letter (4) | OBQA | e-SNLI | SST-2 | Avg. |
|---|---|---|---|---|---|---|---|---|---|---|---|---|
| Prior Best* | $55.0^a$ | $75.3^b$ | $57.4^c$ | $37.9^d$ | - | $91.2^e$ | $73.9^f$ | - | - | - | $97.5^g$ | - |
| *text-davinci-002* | | | | | | | | | | | | |
| Auto-CoT | 47.9 | - | 69.5 | 36.5 | - | 74.4 | 65.4 | 59.7 | - | - | - | - |
| Manual-CoT | 46.9 | 71.3 | 68.9 | 35.8 | 88.8 | 73.5 | 65.4 | 56.6 | 75.5 | 79.1 | 86.2 | 68.0 |
| + Automate-CoT | $49.7\uparrow_{2.8}$ | $74.2\uparrow_{2.9}$ | $73.3\uparrow_{4.4}$ | $37.9\uparrow_{2.1}$ | $90.0\uparrow_{1.2}$ | $76.1\uparrow_{2.6}$ | $67.9\uparrow_{2.5}$ | $58.9\uparrow_{2.3}$ | $79.1\uparrow_{3.6}$ | $82.3\uparrow_{3.2}$ | $87.5\uparrow_{1.3}$ | $70.6\uparrow_{2.6}$ |
| SC | 58.2 | 76.9 | 78.2 | 41.8 | 90.8 | 72.9 | 70.7 | 57.6 | 81.5 | 83.4 | 89.2 | 72.8 |
| + Automate-CoT | $\underline{67.8}\uparrow_{9.6}$ | $\underline{78.9}\uparrow_{2.0}$ | $\underline{80.5}\uparrow_{2.3}$ | $\underline{43.4}\uparrow_{1.6}$ | $\underline{91.9}\uparrow_{1.1}$ | $\underline{80.2}\uparrow_{7.3}$ | $\underline{76.3}\uparrow_{5.6}$ | $\underline{60.8}\uparrow_{3.2}$ | $\underline{84.8}\uparrow_{3.3}$ | $\underline{86.4}\uparrow_{3.0}$ | $\underline{90.6}\uparrow_{1.4}$ | $\underline{76.5}\uparrow_{3.7}$ |
| *code-davinci-002* | | | | | | | | | | | | |
| Auto-CoT | 62.8 | - | - | - | - | - | - | - | - | - | - | - |
| Manual-CoT | 63.1 | 80.4 | 76.4 | 45.3 | 91.8 | 77.9 | 73.2 | 70.4 | 80.4 | 67.5 | 89.7 | 74.2 |
| + Automate-CoT | $67.6\uparrow_{4.5}$ | $83.1\uparrow_{2.7}$ | $78.2\uparrow_{1.8}$ | $47.8\uparrow_{2.5}$ | $92.4\uparrow_{0.6}$ | $81.3\uparrow_{3.4}$ | $75.3\uparrow_{2.1}$ | $75.0\uparrow_{4.6}$ | $83.2\uparrow_{2.8}$ | $71.2\uparrow_{3.7}$ | $90.8\uparrow_{1.1}$ | $76.9\uparrow_{2.7}$ |
| SC | 78.0 | 87.8 | 86.8 | 52.0 | 92.8 | 81.5 | 79.8 | 73.4 | 88.4 | 74.8 | 91.5 | 80.6 |
| + Automate-CoT | $\mathbf{82.4}\uparrow_{4.4}$ | $\mathbf{88.9}\uparrow_{1.1}$ | $\mathbf{87.8}\uparrow_{1.0}$ | $\mathbf{55.6}\uparrow_{3.6}$ | $\mathbf{94.0}\uparrow_{1.2}$ | $\mathbf{84.0}\uparrow_{2.5}$ | $\mathbf{80.6}\uparrow_{0.8}$ | $\mathbf{76.2}\uparrow_{2.8}$ | $\mathbf{89.7}\uparrow_{1.3}$ | $\mathbf{78.3}\uparrow_{3.5}$ | $\mathbf{92.8}\uparrow_{1.3}$ | $\mathbf{82.8}\uparrow_{2.2}$ |

Table 1: The overall performance of Automate-CoT and the comparison against existing models on eleven downstream tasks. Manual-CoT and SC represent chain-of-thought (Wei et al., 2022b) and self-consistency (Wang et al., 2023) methods. **Bold** denotes the best in code-davinci-002-based methods and Underline denotes the best in text-davinci-002-based methods. *: Prior Best is the best performance before CoT comes out. $a$: Cobbe et al. (2021), $b$: Lan et al. (2022), $c$: Pi et al. (2022), $d$: Amini et al. (2019), $e$: Xu et al. (2022), $f$: Chowdhery et al. (2022), $g$: Raffel et al. (2020). Most statistics of Manual-CoT and SC have been obtained directly from their latest version. For some entries they did not report, we obtain the result from DIVERSE (Li et al., 2022b).

**Select:** Both the training and validation sets have a size of 100 to reach a performance and cost trade-off. Then by utilizing the log probability returned by API calls, we calculate the cross-entropy loss of the answer token. Finally, we optimize the latent variables by AdamW (Loshchilov and Hutter, 2019) for 5 epochs with a learning rate of $1 \times 10^{-3}$ and batch size of 10. After optimization, we choose the exemplars combination ($\arg \max \boldsymbol{p}_i$) with the highest validation accuracy to be further evaluated on the test set. By default, we query the language model once to get the answer. Under the self-consistency setting, similar to Wang et al. (2023), we query the language model 40 times and choose the most consistent one as the final answer.

**Hyper-parameter Setting:** Under few-shot setting, we set max_tokens = 256 for all augmentation, selection and inference. In addition, we set logprobs = 5 when training. Moreover, we set temperature = 0.7 for evaluation under self-consistency while temperature = 0 for all other cases.

## 5 Experimental Results

The experimental results are shown in Table 1. We discuss our results in three sections based on the task categories. Automate-CoT are averaged over three runs, and the variance over different runs is reported in Appendix Table 7. Overall, Automate-CoT achieves superior results on all tasks. With text-davinci-002, Automate-CoT outperforms Manual-CoT and SC by 2.6% and 3.7% on average.

With code-davinci-002, Automate-CoT also outperforms Manual-CoT and SC by 2.7% and 2.2%, respectively.

**Arithmetic Reasoning:** For text-davinci-002, Automate-CoT improves Manual-CoT by 2.7% over five arithmetic reasoning tasks. In addition, under the self-consistency setting, Automate-CoT improves SC by a large margin by an average of 3.3%. Moreover, compared to Auto-CoT, Automate-CoT also outperforms it on all three arithmetic tasks (GSM8K, SVAMP, and AQuA). While for code-davinci-002, Automate-CoT achieves an average of 2.4% improvement across all five arithmetic reasoning tasks, illustrating the effectiveness of our proposed approach with different language models. Additionally, Automate-CoT outperforms Auto-CoT in GSM8K by 4.8%, since Auto-CoT only constructs experiments on GSM8K under code-davinci-002. Automate-CoT demonstrates consistent improvement over arithmetic tasks, especially on GSM8K, where it can outperform Manual-CoT by a large margin. Finally, under the self-consistency setting, Automate-CoT also shows similar trends to improve the SC baseline, demonstrating the synergistic effects of our proposed method and self-consistency method.

**Commonsense and Symbolic Reasoning** Similarly, on commonsense and symbolic reasoning tasks, Automate-CoT demonstrates significant improvement over Manual-CoT, SC, and Auto-CoT. It achieves an average of 2.5% and

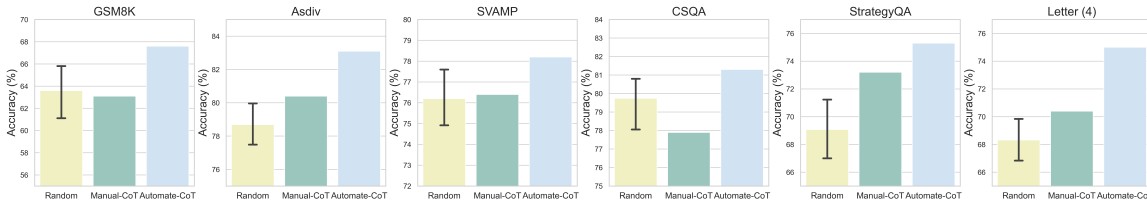

Figure 3: Comparisons between Random Selection, Manual-CoT and Automate-CoT on six datasets.

3.4% improvement on `text-davinci-002` and `code-davinci-002` respectively, demonstrating that our method is effective on different task types. More surprisingly, the improvement in the Letter (4) is significant, demonstrating our method's robustness to deal with out-of-distribution data.

**Non-Reasoning Tasks** Automate-CoT has also reached great success on question answering (OpenBookQA), natural language inference (e-SNLI), and sentiment analysis (SST-2) tasks by an improvement of 2.8%, 3.4% and 1.3%, respectively. The results show that our method can be generalized to various task types and is not limited to reasoning tasks.

## 6 Additional Experiments and Analysis

We further conduct several experiments to evaluate the effectiveness of Automate-CoT and analyze the contributions of each module. Since queries to `text-davinci-002` are limited and expensive, most additional experiments are conducted with `code-davinci-002`.

### 6.1 Effects of Selection Algorithm

After obtaining a large pool of exemplars, a natural question would be what is the performance if we randomly select from the pool regardless of order. In Figure 3, we compare the accuracy obtained by random selection, human-written (Manual-CoT), and our Automate-CoT. For random selection, we randomly sample exemplars from the pool and combine them regardless of order to form the prompts. We repeat this process five times and report the accuracy with an error bar. The results show that random selection suffers from high variance and relatively low accuracy compared to Manual-CoT and Automate-CoT. Surprisingly, we observed the average performance of a random selection from model-generated exemplars can outperform Manual-CoT in some datasets (e.g. GSM8K, CSQA). This also suggests that manual prompt engineering needs to take efforts to design carefully in terms of difficulty,

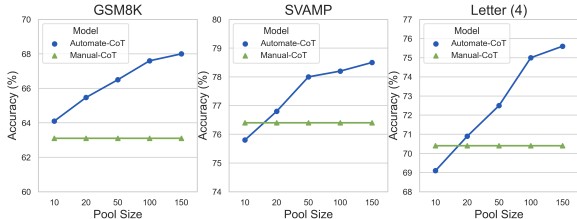

Figure 4: The performance across different pool sizes of Automate-CoT compare with Manual-CoT. Pool size refers to the number of exemplars in the pool.

diversity, and style. In conclusion, if we simply randomly select the exemplars from the pool, it is very likely to obtain a much lower accuracy than the manually written method. However, our Automate-CoT can consistently outperform random selection and Manual-CoT which shows the effectiveness of our method.

### 6.2 Effects of Pool Size

We further conduct a set of experiments to test different pool sizes. As shown in Figure 4, if the pool size is limited to only 10, the performance of Automate-CoT is worse than Manual-CoT or comparable with Manual-CoT. It turns out that if the pool size is small, Automate-CoT is unable to select a good combination to beat carefully designed Manual-CoT. However, Automate-CoT can outperform Manual-CoT when the pool size reaches 20 or larger. The trends show that the performance would be better as pool size keeps increasing. This is intuitive and matches our hypothesis because as pool size increases, there would be more complex, diverse exemplars to choose from. It is expected that the performance would keep increasing, but since more queries for GPT-3 are time-consuming and expensive, we limited these additional experiments to have a max pool size of 150.

### 6.3 Effects of Chain Complexity

It is observed that exemplars written by human are rather simple, so we further explore how chain complexity affect performance. We randomly pick

| RUNS | GSM8K | SVAMP | Letter(4) |
|---|---|---|---|
| Rand(Training Set)$_1$ | 67.55 | 78.2 | 75.0 |
| Rand(Training Set)$_2$ | 67.93 | 77.8 | 76.6 |
| Rand(Training Set)$_3$ | 67.25 | 77.6 | 75.8 |
| Variance | 0.077 | 0.062 | 0.426 |

Table 2: The effect of different randomly chosen training set on performance over three datasets.

| METHOD | GSM8K | SVAMP | Letter (4) |
|---|---|---|---|
| Zero-Shot-CoT | 40.7 | 62.1 | 57.6 |
| Manual-CoT | 46.9 | 73.5 | 56.6 |
| Auto-CoT | 48.9 | 69.5 | **59.7** |
| Zero-Shot-Automate-CoT | **49.1** | **74.3** | 59.3 |
| Automate-CoT | 49.7 | 76.1 | 58.9 |

Table 3: The performance of Automate-CoT in zero-shot setting compared with other baselines. Lightgray highlights our main model which uses a manually constructed chain-of-thought and is not intended for comparison. We list it here only for reference.

8 exemplars with complex rationale chains (each has 9 hops) and refer to them as Complex-CoT. For human-written exemplars (Manual-CoT) Wei et al. (2022b), exemplars are all 2-3 hops. We compare them with our Automate-CoT which has an average hop of 4 and ranges from 2-hop to 6-hop on GSM8K dataset. From Figure 1, Manual-CoT has an overall accuracy of 62%, achieving good results on simple questions. However, it suffers from complex math questions, especially 7-hop and 8-hop questions. Complex-CoT can improve the accuracy on complex questions by a large margin but it performs poorly on simple questions, which only has an overall accuracy of 60%. In contrast, our Automate-CoT can select a combination of different complex-level exemplars automatically. It achieves good results on simple questions and reasonable results on complex questions at the same time, outperforming both Manual-CoT and Complex-CoT by a large margin. The result shows the superiority of our method because it can automatically achieve a complexity-diversity trade-off.

### 6.4 Effects of Training Example Selection

Since training examples to construct CoT are randomly chosen, we also measure the performance vary regarding this random selection. Three different randomly chosen training sets are used to train Automate-CoT and the results are reported in Table 2. According to the result, Automate-CoT shows its robustness to training examples. Randomly chosen training examples have quite a small impact on the result.

### 6.5 Bypass Manual Effort by Zero-shot-CoT

Starting with 4-8 manually constructed chain-of-thought exemplars, our methods show great success in automatically generating, pruning, and selecting suitable exemplars for each task. After that, we raise a new question: *Can we further bypass the effort of writing the initial chain-of-thought exemplars?* Based on current research of Zero-Shot-CoT (Kojima et al., 2022), we found it is possible.

Instead of using 4-8 manual-written exemplars to generate the chains, we simply add *"Let's think step by step."* and let LLMs generate the chains. We test the result under `text-davinci-002` model on GSM8K, SVAMP, and Letter (4) and compare it with Zero-shot-CoT, Manual-CoT and Auto-CoT. Surprisingly, we observe the result can be comparable and even outperform Manual-CoT and Auto-CoT a bit as shown in Table 3. The results further demonstrate that our method can effectively select a suitable combination of exemplars even from a pool that may contain low-quality chains. In conclusion, if a dataset already has manually written chains, our method can be applied to boost the performance. If a dataset does not have manually written chains, our method can still be used to achieve higher accuracy than if it had manually written chains, demonstrating the superiority of our method.

## 7 Ablation Study

In this section, we further conduct ablation experiments to verify the advantage of the generated prompts on four factors, respectively.

**Advantage over Order Factor** The advantages of Automate-CoT on order factor can be viewed in two ways. Firstly, it requires a large human effort to determine a good order by trying many different orders on validation sets. However, Automate-CoT can automatically construct the exemplars without further adjustment to have a good result. Secondly, Automate-CoT is less affected by the order sensitivity. We further conduct an experiment to compare selected exemplars and random permutations of Automate-CoT's selected exemplars as shown in Table 4. We randomly permutate the selected exemplars to see how performance varies compared to the selected order by Automate-CoT. It is observed that the order sensitivity still exists and our se-

| RUNS | GSM8K | SVAMP | Letter(4) |
|---|---|---|---|
| perm(Automate-CoT)$_1$ | 66.7 | 77.2 | 73.0 |
| perm(Automate-CoT)$_2$ | 66.6 | 78.4 | 72.6 |
| perm(Automate-CoT)$_3$ | 66.9 | 78.0 | 72.0 |
| perm(Automate-CoT)$_4$ | 67.8 | 78.2 | 74.2 |
| perm(Automate-CoT)$_5$ | 67.5 | 78.1 | 75.0 |
| Automate-CoT | **68.4** | **78.7** | **75.2** |
| Mean$_{\pm std}$ | 67.3$_{\pm 0.64}$ | 78.2$_{\pm 0.46}$ | 73.7$_{\pm 1.21}$ |

Table 4: Comparison of different permutations orders of Automate-CoT's selected examplars.

lected exemplars have better performance than that of all 5 random permutation runs, demonstrating Automate-CoT can automatically choose a good order without any human effort.

**Advantage over Complexity Factor** As discussed in the complexity factor of Section 2, we show that the complexity of manually written chains is quite simple (less than or equal to 3 hops). It would require more human effort to design complex rationales. However, Automate-CoT can automatically augment and select examples with different complexity, reaching a better trade-off accuracy between simple questions and complex questions (Appendix Table 9).

**Advantage over Diversity Factor** The diversity of Manual-CoT or Complexity-CoT is limited. For example, every exemplar of Complexity-CoT has the same complexity and every exemplar of Manual-CoT ranges from 1-3 hops as illustrated in the motivation section. However, Automate-CoT can automatically select an optimal combination of complexity that best suits the dataset. For example, our selected exemplars on GSM8K have an average hop of 5.4 and range from 3 hops to 8 hops as shown in Appendix G. It contains both simple exemplars and complex exemplars which reach the best performance.

**Advantage over Style Factor** Our extensive experience with multiple experiments indicates that a good linguistic style is typically formal and detailed. This style entails the use of (1) explicit and logical connection words (e.g., "so", "that means"), (2) detailed reasoning steps within a single sentence, (3) symbols when appropriate (e.g., using the $ symbol to denote monetary values), and (4) minimizing the use of abbreviations. We further conduct an ablation experiment to test how our method can choose the examples with better style. Firstly, we use Automate-CoT to select 8 rationale exem-

plars $S_1 = [A_1, B_1, C_1, D_1, E_1, F_1, G_1, H_1]$ for GSM8K. Then we copy this set and edit its written / linguistic style manually to be worse while keeping the order, complexity, and diversity the same which gives $S_2 = [A_2, B_2, C_2, D_2, E_2, F_2, G_2, H_2]$. Now we have 16 examplars says $S = [A_1, A_2, B_1, B_2, ..., H_1, H_2]$. A-H represents the No.1-8 exemplars. Subscript 1 represents the originally selected exemplars and 2 represents the edited ones. Then, Automate-CoT selects 8 exemplars from the previous 16 exemplars. Note that we limit Automate-CoT to select exactly one of $[A_1, A_2]$ and $[B_1, B_2]$ ... and keep the same order A-H. Subsequently, when we perform Automate-CoT algorithm, we observe that Automate-CoT is able to successfully select the original exemplars $S_1$. Furthermore, we find that the selected exemplars can outperform the non-selected exemplars by 2%.

# 8 Related Work

In this section, we first review the recent progress of prompt-based learning (§8.1) and chain-of-thought prompting (§8.2), and then discuss the black-box optimization methods (§8.3).

## 8.1 Prompt-based Learning

Prompt-based Learning (Prompting) aims to leverage large language models (LLMs) (Devlin et al., 2018; Liu et al., 2019; He et al., 2021; Diao et al., 2020, 2021) to trigger helpful knowledge for downstream tasks. Existing prompting methods can be categorized into two types based on their nature: 1) discrete prompts (Wallace et al., 2019; Shin et al., 2020; Jiang et al., 2020; Yuan et al., 2021; Haviv et al., 2021; Gao et al., 2021; Ben-David et al., 2022; Davison et al., 2019; Su et al., 2022; Diao et al., 2022; Guo et al., 2023) and continuous prompts (Zhong et al., 2021; Qin and Eisner, 2021; Hambardzumyan et al., 2021; Liu et al., 2021; Han et al., 2021; Li and Liang, 2021; Yang et al., 2023). Discrete prompts optimize a sequence of discrete tokens, while continuous prompts optimize a sequence of vectors. One of the most important advantages of prompting is saving fine-tuning costs by refraining from the parameter changes of large language models, and we only need to optimize a small set of parameters.

## 8.2 Chain-of-thought Prompting

Chain-of-thought (Wei et al., 2022b) introduces a chain of rationale steps for each exemplar of

in-context learning and significantly improves the performance on several complex tasks like arithmetic reasoning, commonsense reasoning, and symbolic reasoning. Based on this simple yet effective idea, many following works propose different strategies to improve it: self-consistency (Wang et al., 2023), explanation learning (Lampinen et al., 2022), complexity-based prompting (Fu et al., 2023), self-training (Huang et al., 2022), voting verifier (Li et al., 2022a), zero-shot prompting (Kojima et al., 2022; Fung et al., 2022), and bootstrapping (Zelikman et al., 2022).

### 8.3 Black-box Optimization

Nowadays, large language models provide services as commercial APIs deployed in the cloud, such as OpenAI's GPT-3 (Brown et al., 2020) and ChatGPT[3]. It usually accepts query inputs and outputs the predictions with a web interface. Their model parameters and gradients are not accessible, causing difficulties in optimization with gradients. Previous research on black-box optimization mainly focuses on score-based black-box adversarial attack (Ilyas et al., 2018, 2019; Huang and Zhang, 2020; Andriushchenko et al., 2020; Cheng et al., 2019). Most recently, black-box prompt learning (Diao et al., 2022; Sun et al., 2022; Prasad et al., 2022) is introduced, aiming to optimize the prompts without accessing gradients, but their models suffer from limited reasoning abilities and are limited to zero-shot settings with classification task.

## 9   Conclusion

In this paper, we proposed a chain-of-thought optimization method consisting of three steps: augment, prune, and select. Automate-CoT first generates rationale chains according to the standard CoT process with several exemplars, and then prunes those incorrect ones according to the consistency of the predicted answer and ground-truth answer. Finally, we apply a variance-reduced policy gradient strategy to estimate the gradients and optimize the latent variables to select better CoTs. Experimental results demonstrate the effectiveness of our method on arithmetic reasoning, commonsense reasoning, symbolic reasoning tasks, and non-reasoning tasks.

## 10   Limitations

It is shown that Automate-CoT demonstrates superior performance over previous chain-of-thought

prompting methods. However, despite these exciting results, there are still some limitations to our current work, as well as potential opportunities for future research.

**Comparision with Fine-tuning** : Our main baselines include original chain-of-thought (Wei et al., 2022b), self-consistency (Wang et al., 2023) which are manual-written based prompt method. In addition, we also compare the clustering-based and retrieval-based methods to select the prompt exemplars like Auto-CoT (Zhang et al., 2023), BM25 (Robertson, 2009), PromptPG (Lu et al., 2023). As large language models are dominating the field, the performance of training the large language models by using these labeled data might be interesting. However, it is not covered in this study due to the prompt setting of this study and limited resources.

**Prompt Style Definition** : Another limitation of this work is that it does not provide a rigorous definition of what constitutes good versus bad linguistic style. While we have observed several patterns of good and bad style during numerous experiments, and the results show that Automate-CoT is able to mitigate style sensitivity in Manual-CoT, we cannot determine what perfect style entails. As such, we acknowledge that defining what constitutes good versus bad linguistic style can be a challenging task and an important area for further exploration and development.

## Acknowledgments

We thank the anonymous reviewers for their valuable suggestions. This work was supported by the General Research Fund (GRF) of Hong Kong (No. 16310222). Shizhe Diao was supported by the Hong Kong Ph.D. Fellowship Scheme (HKPFS).

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

---

**Algorithm 1** The black-box optimization procedures.

---

**Require:** Input batch $S$, Label batch $Y$, Parameter of categorical distribution $\boldsymbol{p}_1, \cdots, \boldsymbol{p}_n$, Prediction model $\mathcal{G}$, Loss function $\mathcal{L}$.

1: **for** $k \leq I$ **do**
2:     Sample $j_1^{(k)} \sim \mathrm{Cat}(\boldsymbol{p}_1), \cdots, j_n^{(k)} \sim \mathrm{Cat}(\boldsymbol{p}_n)$
3:     $T^{(k)} = t_1^{(k)} \cdots t_n^{(k)} = \mathcal{V}[j_1^{(k)}] \cdots \mathcal{V}[j_n^{(k)}]$
4: **end for**
5: $\mathcal{L}_{\mathrm{avg}} = \frac{1}{I} \sum_{k=1}^{I} \mathcal{L}(\mathcal{G}[T^{(k)}, S], Y)$
6: **for** $i \leq n$ **do**
7:     $\boldsymbol{g}_{\boldsymbol{p}_i}^{vr} = \frac{1}{I-1} \sum_{k=1}^{I} \nabla_{\boldsymbol{p}_i} \log P(t_i^{(k)})(\mathcal{L}(\mathcal{G}[T^{(k)}, S], Y) - \mathcal{L}_{\mathrm{avg}})$
8:     $\boldsymbol{p}_i \leftarrow \mathrm{proj}_{\mathcal{C}}(\boldsymbol{p}_i - \eta \cdot \boldsymbol{g}_{\boldsymbol{p}_i}^{vr})$
9: **end for**
10: **return** $\boldsymbol{p}_1, \cdots \boldsymbol{p}_n$

---

## A  Algorithm Details

In this section, we provide more details about the derivation of the equation (1) in Section 3.2. Given the loss function:

$$\mathbb{E}_T\left[\mathcal{L}(T)\right] = \int \mathcal{L}(T)P(T)\,\mathrm{d}T \tag{4}$$

We can estimate the gradient of $\boldsymbol{p}_i$ by:

$$
\begin{aligned}
\nabla_{\boldsymbol{p}_i}\mathbb{E}_T\left[\mathcal{L}(T)\right] &= \int \mathcal{L}(T)\nabla_{\boldsymbol{p}_i}P(T)\,\mathrm{d}T \\
&= \int \mathcal{L}(T)\frac{P(T)}{P(T)}\nabla_{\boldsymbol{p}_i}P(T)\,\mathrm{d}T \\
&= \int P(T)\mathcal{L}(T)\nabla_{\boldsymbol{p}_i}\log P(T)\,\mathrm{d}T \\
&= \mathbb{E}_{P(T)}\left[\mathcal{L}(T)\nabla_{\boldsymbol{p}_i}\log \Pi_{j=1}^n P(t_j)\right] \\
&= \mathbb{E}_{P(T)}\left[\mathcal{L}(T)\nabla_{\boldsymbol{p}_i}\sum_{j=1}^n \log P(t_j)\right] \\
&= \mathbb{E}_{P(T)}\left[\mathcal{L}(T)\nabla_{\boldsymbol{p}_i}\log P(t_i)\right]
\end{aligned}
\tag{5}
$$

The $j$-th component of $\nabla_{\boldsymbol{p}_i}\log P(t_i)$ could be solved explicitly by:

$$\nabla_{p_{i,j}}\log P(t_i) = \nabla_{p_{i,j}}\log p_{i,j_i} \tag{6}$$

When $j = j_i$, it is obvious that $\nabla_{p_{i,j}}\log P(t_i) = \frac{1}{p_{i,j_i}}$. When $j \neq j_i$, equation (6) is calculated by:

$$
\begin{aligned}
\nabla_{p_{i,j}}\log P(t_i) &= \nabla_{p_{i,j}}\log(1 - \sum_{k=1,k\neq j_i}^N p_{i,k}) \\
&= -\frac{1}{1 - \sum_{k=1,k\neq j_i}^N p_{i,k}} \\
&= -\frac{1}{p_{i,j_i}}
\end{aligned}
\tag{7}
$$

Therefore, we adopted a variance-reduced policy gradient estimator (VR-PGE) as described in Williams (1992); Dong et al. (2020); Zhou et al. (2021) to mitigate the high-variance issue of PGE. The estimated gradient is calculated by:

$$g_{p_i}^{vr} = \frac{1}{I-1} \sum_{k=1}^{I} \left( \mathcal{L}(T^{(k)}) - \frac{1}{I} \sum_{j=1}^{I} \mathcal{L}(T^{(j)}) \right) \nabla_{p_i} \log P(t_i) \tag{8}$$

where $T^{(k)}, k = 1, \cdots, I$ are sampled independently from $P(T)$.

Thus, the prompt token distribution $p_i$ can be updated by a projected stochastic gradient descent algorithm:

$$p_i \leftarrow \text{proj}_{\mathcal{C}}(p_i - \eta \cdot g_{p_i}^{vr}), i = 1, \cdots, n \tag{9}$$

where $\eta$ is the learning rate of prompt learning, $I$ is the sample size, and $\text{proj}_{\mathcal{C}}$ is the projection calculation. The detailed training procedure of our VR-PGE algorithm is displayed in Algorithm 1.

## B  Detailed Experimental Setting

| DATASET | TASK TYPE | # EX. | # EVAL. | EVAL. SPLIT | TRANSFERRED |
|---|---|---|---|---|---|
| GSM8K (Cobbe et al., 2021) | Arithmetic | 8 | 1319 | Test | ✗ |
| ASDiv (Miao et al., 2020) | Arithmetic | 8 | 2096 | Test | ✓ |
| SVAMP (Patel et al., 2021) | Arithmetic | 8 | 1000 | Test | ✓ |
| AQuA (Ling et al., 2017) | Arithmetic | 4 | 254 | Test | ✗ |
| SingleOp♣ | Arithmetic | 8 | 562 | Test | ✓ |
| CSQA♦ (Talmor et al., 2019) | Commonsense | 7 | 1221 | Validation | ✗ |
| StrategyQA♦ (Geva et al., 2021) | Commonsense | 6 | 1880 | Validation | ✗ |
| Letter (4) (Wei et al., 2022b) | Symbolic | 4 | 500 | Test (OOD) | ✗ |
| OpenBookQA (Mihaylov et al., 2018) | Question Answering | 4 | 500 | Test | ✗ |
| e-SNLI♥ (Camburu et al., 2018) | Narural Language Inference | 6 | 1000 | Test | ✗ |
| SST-2♦ (Socher et al., 2013) | Sentiment Analysis | 6 | 872 | Validation | ✗ |

Table 5: The overall statistics of the datasets. # EX.: the number of few-shot chain-of-thought exemplars used to prompt each task. # EVAL.: the number of evaluation data. EVAL. SPLIT: evaluation split. TRANSFERRED: a checkmark means that the exemplars are generated and trained from other datasets and then applied to this task. ♣: SingleOp is a subset of MAWPS (Koncel-Kedziorski et al., 2016). ♦: CSQA, StrategyQA, and SST-2 do not have publicly available test set labels, so we simply follow the setting by Wei et al. (2022b) and Wang et al. (2022) to evaluate the performance of the validation set. ♥: Following Wang et al. (2022), we evaluate the first 1,000 data points for a fair comparison.

### B.1  Datasets and Evaluation Metrics

Following Wei et al. (2022b), we conduct our experiments on eight reasoning tasks, including five math word problem datasets: GSM8K, ASDiv, SVAMP, AQuA, and SingleOp; two commonsense reasoning datasets: CommonsenseQA (CSQA) and StrategyQA, and one symbolic reasoning task: Last Letter Concatenation (Letter (4)). We also generalize our method to non-reasoning tasks including one question-answering task (OpenBookQA), one natural language inference task (e-SNLI), and one sentiment analysis task (SST-2). The detailed statistics of the datasets are listed in Table 5.

To make a fair comparison with our baselines, we use the same number of exemplars as Wei et al. (2022b) and Wang et al. (2022), as shown in Table 5. We keep the same setting for the evaluation split as well. By default, we use the test split for evaluation, and for datasets that do not have publicly available test set labels, we evaluate the validation set instead. In addition, for last letter concatenation, since the model has already achieved almost 100% accuracy under the in-distribution setting, we only test the out-of-distribution (OOD) setting, Letter (4), where prompts are 2-letters, and test examples are 4-letters. The evaluation metric for all tasks is the exact match accuracy. First, we conduct pre-processing for predictions to remove all the special symbols. For example, "$100,000" will be processed to "100000". Then we check if it has the same value as the ground truth to calculate the exact match accuracy.

### B.2  Baselines

In our experiments, the following three methods serve as the main baselines:

- chain-of-thought (Manual-CoT) (Wei et al., 2022b): standard chain-of-thought prompting which provides manual-written intermediate reasoning steps.

- self-consistency (SC) (Wang et al., 2023): an improved version of CoT. Instead of greedy decoding, it samples a diverse set of reasoning paths and chooses the most common answer.
- Auto-CoT (Zhang et al., 2023): an automatic exemplars construction method that applies clustering techniques to sample questions and then generates chains.

Our experiments are conducted with two popular large language models:

- GPT-3 (Brown et al., 2020): we test an advanced version of GPT-3, `text-davinci-002`, which corresponds to InstructGPT (Ouyang et al., 2022) model.
- CodeX (Chen et al., 2021): we test `code-davinci-002` which has better code representation ability.

We utilize the public APIs directly from OpenAI's services[4]. In our main experiments, we test on both `text-davinci-002` and `code-davinci-002` engines. However, in additional experiments, we mainly test on `code-davinci-002` for two reasons : (1) It is the most capable model available at the time we were conducting our experiments, consistent with the observations in previous studies (Wei et al., 2022b; Wang et al., 2023; Miao et al., 2020). (2) Compared to costly `text-davinci-002`, it is free of charge because we are in the initial limited beta period during our experiments process.

## B.3  Implementation

**Augment and Prune:** Following Wei et al. (2022b) and Wang et al. (2022), we keep the same number of exemplars (4-8) listed in Table 5. For main experiments, we augment and prune a pool of 100 high-quality exemplars for all datasets. Firstly, pool construction questions are randomly sampled and then fed to LLMs to construct model-generated answers with rationale chains. Given that some datasets only have the test split, we use the pool result of GSM8K and transferred it to these datasets for further inference. Here for arithmetic reasoning tasks, pool construction questions are randomly sampled from the training split of GSM8K and AQuA. For CSQA and StrategyQA, exemplars are randomly sampled from the official training split (Talmor et al., 2019) and question-only set from BIG-bench collaboration (Srivastava et al., 2022). For letter concatenation, exemplars are randomly sampled from the 2-letter set. After the pool is constructed, we use labels to prune the incorrect model-generated exemplars and retain 100 high-quality exemplars.

**Select:** The train set and validation set are also randomly sampled following the same rule as above except Letter (4) dataset. Since LLM has already reached almost 100% accuracy on the 2-letter set, we choose to optimize the model based on the 3-letter OOD set. Thus the train set and validation set are randomly sampled from the 3-letter set. Both the train and validation sets have a size of 100 to reach a performance and cost trade-off. Then by utilizing the log probability returned by API calls, we calculate the cross-entropy loss of the answer token. Finally, we optimize the latent variables by AdamW (Loshchilov and Hutter, 2019) for 5 epochs with a learning rate of $1 \times 10^{-3}$ and batch size of 10. After optimization, as shown in Figure 2 inference stage, we choose the exemplars combination ($\arg\max \boldsymbol{p}_i$) with the highest validation accuracy to be further evaluated on the test set. By default, we query the language model once to get the answer. Under the self-consistency setting, similar to Wang et al. (2023), we query the language model 40 times and choose the most consistent one as the final answer.

**Hyper-parameter Setting:** Under few-shot setting, we set max_tokens = 256 for all augmentation, selection and inference. In addition, we set logprobs = 5 when training. Moreover, we set temperature = 0.7 for evaluation under self-consistency while temperature = 0 for all other cases. Under zero-shot setting (§6.5), we keep the same hyper-parameters as Kojima et al. (2022) which first uses max_tokens = 128 for generating the rationale chains and then uses max_tokens = 32 for generating the answers to construct the pool. The hyper-parameters for selecting and evaluating are the same as the few-shot setting above.

## C  More Experiment Results

## C.1  Experiments under ChatGPT

To further verify the effectiveness of Automate-CoT, we further conduct the experiments on gpt-3.5-turbo. Automate-CoT also shows consistent improvement on each task with 2.8% improvement on arithmetic

---

[4]https://openai.com/api/

| METHOD | GSM8K | ASDIV | SVAMP | AQUA | SINGLEOP | CSQA | STQA | LETTER (4) | OBQA | e-SNLI | SST-2 | AVG. |
|---|---|---|---|---|---|---|---|---|---|---|---|---|
| Manual-CoT | 63.1 | 77.1 | 78.1 | 44.9 | 90.0 | 77.5 | 59.7 | 73.0 | 80.0 | 80.9 | 85.3 | 73.6 |
| + BM25 | 64.2 | 73.7 | 73.8 | 45.3 | 87.9 | 76.1 | 58.9 | 73.4 | 81.4 | 76.3 | 87.2 | 72.6 |
| + PromptPG | 66.6 | 76.7 | 75.6 | 46.1 | 89.1 | 77.8 | 60.2 | 74.8 | 81.8 | 77.8 | 87.8 | 74.0 |
| + K-Means | 66.4 | 76.6 | 77.6 | 45.7 | 89.7 | 79.0 | 60.0 | 73.6 | 80.4 | 78.4 | 84.1 | 73.8 |
| + Automate-CoT | **68.0**↑$_{4.9}$ | **81.7**↑$_{4.6}$ | **79.1**↑$_{1.0}$ | **46.9**↑$_{2.0}$ | **91.5**↑$_{1.5}$ | **80.5**↑$_{3.0}$ | **64.5**↑$_{4.8}$ | **76.2**↑$_{3.2}$ | **83.0**↑$_{3.0}$ | **81.4**↑$_{0.5}$ | **87.7**↑$_{2.4}$ | **76.4**↑$_{2.8}$ |

Table 6: The overall performance of Automate-CoT under gpt-3.5-turbo and the comparison with retrieval-based and clustering-based exemplars selection methods.

| METHOD | GSM8K | ASDIV | SVAMP | AQUA | SINGLEOP | CSQA | STQA | LETTER (4) |
|---|---|---|---|---|---|---|---|---|
| | | | | *text-davinci-002* | | | | |
| Automate-CoT | 0.14 | 0.29 | 0.17 | 0.21 | 0.08 | 0.06 | 0.26 | 0.04 |
| Automate-CoT(SC) | 0.02 | 0.18 | 0.06 | 0.14 | 0.04 | 0.01 | 0.07 | 0.04 |
| | | | | *code-davinci-002* | | | | |
| Automate-CoT | 0.19 | 0.78 | 0.33 | 0.09 | 0.05 | 0.17 | 0.95 | 0.02 |
| Automate-CoT(SC) | 0.09 | 0.09 | 0.13 | 0.01 | 0.06 | 0.03 | 0.09 | 0.08 |

Table 7: The variance of the results in Table 1 over 3 runs. (SC) denotes under self-consistency setting.

reasoning, 3.9% improvement on commonsense reasoning, 3.2% on symbolic reasoning, and 2.8% improvement overall as shown in Table 6.

## C.2 Comparison with Retrieval Methods

We also compare Automate-CoT with simple retrieval method BM25 (Robertson, 2009) and reinforcement learning-based retrieval method PromptPG (Lu et al., 2023). We first implemented a BM25 selection method and tested the performance on all the datasets. The results are shown in Table 6. It indicates that retrieval-based methods can only select examples with similar meaning to the query question while the diversity is overlooked. As shown in the table, the average performance of the BM25 retrieval-based method even has a 1% degradation compared to Manual-CoT, and 3.8% lower than Automate-CoT. A similar phenomenon is observed in Auto-CoT (Zhang et al., 2023), which indicates that with similar questions being sampled for test questions, Retrieval-Q-CoT is negatively affected by misleading by similarity.

In addition, we also compare with PromptPG (Lu et al., 2023), a dynamic example-selection baseline. We adopt the same setting as ours for PomptPG, where the number of training examples is 100, the size of the candidate pool is 100, and the backbone model is gpt-3.5-turbo. Further, we keep the same prompt format as the original chain-of-thought and ours. The other settings we use are consistent with the settings provided by their original code. The results are shown in Table 6. It indicates that Automate-CoT outperforms PromptPG.

## C.3 Comparison with Clustering Methods

We further conduct additional experiments to compare Automate-CoT with methods selecting demonstration exemplars through clustering. We use K-Means as the clustering method and create k clusters according to the number of exemplars specified in Table 5. Then we use these $k$ representative exemplars as the demonstration exemplars to prompt the language models. The results are shown in Table 6. It indicates that clustering-based methods can select examples with different semantic meanings and generally perform better than Manual-CoT. However, the complexity and diversity are overlooked. For example, most of the selected few-shot exemplars in GSM8K have around 3-4 hops where complex questions and moderately difficult questions are overlooked. As a result, it generally performs worse than Automate-CoT with a 2.6% gap.

## C.4 Variance Report

Since Automate-CoT's results in Table 1 are averaged over three runs, we also report the variance in Table 7 here. It is observed that Automate-CoT achieves quite a low variance, especially compared to the large variance of Manual-CoT as shown in § 2 Motivation.

# D  Additional Comparison with Fine-tuning

Since our method uses a training-based pipeline, we also compare it with fine-tuning large language models in terms of the number of parameters, training cost, estimated total training cost, and required training set size. As shown in the study of Cobbe et al. (2021), fine-tuning on gpt-3 requires thousands (e.g., 8000) of training examples to be effective while Automate-CoT only needs 100 training examples. In addition, fine-tuning has a larger training and inference cost than Automate-CoT because it not only requires a one-off fine-tuning cost but also has a higher unit price on subsequent usage.

For Automate-CoT, under the setting of gpt-3.5-turbo, the direct usage is \$ 0.0015 / 1k tokens for input and \$ 0.002 / 1k tokens for output. With the training epochs of 3, a training set size of 100 and a validation set size of 100, an input length of around 750 tokens and an average output length of 150 tokens, it takes about $(750/1000 \cdot 0.0015 + 150/1000 \cdot 0.002) \cdot 100 \cdot 10 \cdot 3 + (750/1000 \cdot 0.0015 + 150/1000 \cdot 0.002) \cdot 100 \cdot 3 =$ \$ 4.7. However, for fine-tuneing, given the training price of gpt-3.5-turbo is \$ 0.008 / 1K tokens, the usage of finetuned gpt-3.5-turbo is \$ 0.0015 / 1K tokens for input and \$ 0.002 / 1K tokens for output tokens. Under the finetuning setting, suppose the average length of training examples is 300 tokens, and training a whole training set of 8000 examples for 3 epochs takes about $300/1000 \cdot 8000 \cdot 3 \cdot 0.008 =$ \$ 57.6, which costs 12x more than Automate-CoT.

It is also worth noting that the further usage of finetuned gpt-3.5-turbo is \$ 0.012 / 1K tokens for input and \$ 0.016 / 1K tokens for output while Automate-CoT remains the normal cost, which is 8x less cost than fine-tuning.

| METHOD | # of Training Params | Cost | Est. Total Cost | Train Set Size |
|---|---|---|---|---|
| Fine-tuning | Unknown

but should $\geq$ 175B | \$0.008/1K tokens (Train)

\$0.012/1K tokens (Input Usage)
\$0.016/1K tokens (Output Usage) | \$ 9.1
\$ 12.7
\$ 20.0
\$ 34.3
\$ 63.1 | 500
1000
2000
4000
8000 |
| Automate-CoT | # of exemplars $\times$ Pool Size | \$0.0015/1K tokens (Input Usage)
\$0.002/1K tokens (Output Usage) | \$ 6.6 | 100 |

Table 8: Comparison between Fine-tuning and Automate-CoT on GSM8K. The cost is copied from the OpenAI official website. [5]

# E  Additional Analysis

We list some additional analysis here that cannot be put in the main section because of the page limit.

## E.1  Effects of Several Tricks

Previous studies have found some tricks like add *"Let's think step by step."* before each rationale chain and replace "Q:" with "Question:" (Fu et al., 2023; Kojima et al., 2022) can boost the performance on top of Manual-CoT. Following their settings, we also test Automate-CoT with tricks on GSM8K as an additional experiment. By adding tricks, Automate-CoT can further boost the accuracy to 69.8% (+2.2%) under the normal setting and 83.0% (+0.6%) under the self-consistency setting, respectively.

# F  Exact Match Number over Each Hop

The exact match number over each hop of Figure 1 is reported in Table 9.

---

[5] https://openai.com/pricing

| HOP | Total Number | Manual-CoT | Automate-CoT | Complex-CoT |
|---|---|---|---|---|
| 2 | 326 | 269 = 82.52% | **272 = 83.44%** | 227 = 69.63% |
| 3 | 370 | 265 = 71.62% | **268 = 72.44%** | 232 = 62.70% |
| 4 | 298 | 159 = 53.35% | **196 = 65.77%** | 166 = 55.70% |
| 5 | 174 | 83 = 47.70% | **110 = 63.21%** | 98 = 56.32% |
| 6 | 88 | 26 = 29.54% | 37 = 42.05% | **39 = 44.32%** |
| 7 | 40 | 10 = 25% | 12 = 30% | **16 = 40%** |
| 8 | 20 | 3 = 15% | 4 = 20% | **5 = 25%** |
| 9 | 2 | 0 | 0 | 0 |
| 11 | 1 | 0 | 0 | **1** |
| Overall | 1319 | 815 = 61.8% | **899 = 68.2%** | 784 = 59.4% |

Table 9: The exact match number across the different numbers of hops on GSM8K. **Bold** represents the best among each hop. The percentage accuracy is calculated for each hop.

# G  Full Exemplars generated by Automate-CoT

| DATASET | Exemplars | Hops |
|---|---|---|
| GSM8K | Q: Ralph is going to practice playing tennis with a tennis ball machine that shoots out tennis balls for Ralph to hit. He loads up the machine with 175 tennis balls to start with. Out of the first 100 balls, he manages to hit 2/5 of them. Of the next 75 tennis balls, he manages to hit 1/3 of them. Out of all the tennis balls, how many did Ralph not hit?
A: Ralph started with 175 tennis balls. He hit 2/5 of the first 100 balls, so he hit 2/5 * 100 = 40 balls. He hit 1/3 of the next 75 balls, so he hit 1/3 * 75 = 25 balls. In total he hit 40 + 25 = 65 balls. He did not hit 175 - 65 = 110 balls. The answer is 110. | 4-hop |
| | Q: Hans booked a room in a hotel. The hotel has 10 floors with 10 identical rooms on each floor. Because of an accident, the last floor is unavailable for the guests. Considering there are no other guests, in how many different rooms could Hans be checked in?
A: There are 10 floors with 10 rooms each. The last floor is unavailable. So there are 9 * 10 = 90 rooms available. The answer is 90. | 3-hop |
| | Q: James earns $20 an hour while working at his main job. He earns 20% less while working his second job. He works 30 hours at his main job and half that much at his second job. How much does he earn per week?
A: James earns $20 an hour at his main job. He earns 20% less at his second job. That means he earns $20 - 20% of $20 = $20 - $4 = $16 an hour at his second job. He works 30 hours at his main job and half that much at his second job. That means he works 30 / 2 = 15 hours at his second job. In total he works 30 + 15 = 45 hours. He earns $20 an hour at his main job and $16 an hour at his second job. In total he earns $20 * 30 + $16 * 15 = $600 + $240 = $840. The answer is 840. | 8-hop |
| | Q: Leah earned $28 working odd jobs around the neighborhood. She spent a seventh of it on a milkshake and put half of the rest in her savings account. She left the remaining money in her wallet. Her dog got ahold of her wallet and shredded all the money inside but $1. How many dollars did Leah lose?
A: Leah earned $28. She spent a seventh of it on a milkshake, so she spent $28 / 7 = $4 on the milkshake. She put half of the rest in her savings account, so she put $28 - $4 / 2 = $12 in her savings account. She left the remaining money in her wallet. She lost all the money in her wallet except $1. So she lost $28 - $4 - $12 - $1 =$11. The answer is 11. | 6-hop |
| | Q: Sam and Jeff had a skipping competition at recess. The competition was split into four rounds. Sam completed 1 more skip than Jeff in the first round. Jeff skipped 3 fewer times than Sam in the second round. Jeff skipped 4 more times than Sam in the third round. Jeff got tired and only completed half the number of skips as Sam in the last round. If Sam skipped 16 times in each round, what is the average number of skips per round completed by Jeff?
A: Sam skipped 16 times in each round. In the first round, Jeff skipped 16 - 1 = 15 times. In the second round, Jeff skipped 16 - 3 = 13 times. In the third round, Jeff skipped 16 + 4 = 20 times. In the last round, Jeff skipped 16 / 2 = 8 times. The average number of skips per round completed by Jeff is (15 + 13 + 20 + 8) / 4 = 56 / 4 = 14. The answer is 14. | 6-hop |
| | Q: A Statistics student wants to find out the average daily allowance of the middle school students. According to his survey, 2/3 of the students receive an average of $6 allowance per day while the rest gets an average of $4 a day. If he surveyed 60 students, what is the total amount of money those 60 students get in a day?
A: 2/3 of the students receive $6 a day. The rest receive $4 a day. The total number of students is 60. So, 2/3 of 60 students receive $6 a day. The rest of the students, which is 60 - 2/3 * 60 = 20 students, receive $4 a day. The total amount of money those 60 students get in a day is 2/3 * 60 * $6 + 20 * $4 = $240 + $80 = $320. The answer is 320. | 6-hop |
| | Q: Ali had a collection of seashells. He started with 180 seashells. He then gave away 40 seashells to his friends. He also gave 30 seashells to his brothers. If he sold half of the remaining seashells, how many seashells did he have left?
A: He started with 180 seashells. He gave away 40 to his friends and 30 to his brothers. That means he gave away 40 + 30 = 70 seashells. He has 180 - 70 = 110 seashells left. He sold half of them, so he has 110 / 2 = 55 seashells left. The answer is 55. | 5-hop |
| | Q: Marvin and Tina were selling candy bars to help fund their class trip. The candy bars cost $2 each. Marvin sold 35 candy bars total. Tina sold three times the number of candy bars as Marvin. How much more money did Tina make for the class trip selling candy bars compared to Marvin?
A: Marvin sold 35 candy bars. Tina sold 3 times as many, so she sold 35 * 3 = 105 candy bars. Each candy bar cost $2, so Marvin made 35 * $2 = $70. Tina made 105 * $2 = $210. The difference is $210 - $70 = $140. The answer is 140. | 5-hop |

Table 10: One example of selected model-generated exemplars with rationale chains of average hops = 5.4. This set of exemplars is trained and selected on GSM8K and transferred to other arithmetic reasoning tasks.

| DATASET | Exemplars |
|---|---|
| AQuA | Q: If Tim had lunch at $50 and he gave 20% tip, how much did he spend? Answer Choices: (a) $60.00 (b) $35.42 (c) $60.60 (d) $21.56 (e) $78.45 
 A: The tip is 20% of what he paid for lunch. tip = 20% of 50.00 = (20/100)*50.00 = = $10.00. Total spent 50.00 + 10.00 = $60.00. The answer is (a). 

 Q: A person can walk at a constant rate of 8mph and can bike at a rate of 16mph. If he wants to travel 64 miles in 8 hours using bike and walking at their constant rates, how much distance would he require to walk? Answer Choices: (a) 20 (b) 30 (c) 48 (d) 64 (e) 72 
 A: Total distance = 64. Distance = Speed * Time. Walking speed = s1 = 8. Walking time = t1. Bike speed = s2 = 16. Time traveled in bike = t2. d1 + d2 = 64. s1t1 + s2t2 = 64. 8*t1 + 16*t2 = 64. t1 + 2*t2 = 8 —– (1). Given: t1 + t2 = 8 —– (2). (1) - (2) −− > t2 = 0 and t1 = 8 - 0 = 8. Walking distance = s1*t1 = 8*8 = 64. The answer is (d). 

 Q: The output of a factory was increased by 10% to keep up with rising demand. To handle the holiday rush, this new output was increased by 20%. By approximately what percent would the output now have to be decreased in order to restore the original output? Answer Choices: (a) 20% (b) 24% (c) 30% (d) 32% (e) 79% 
 A: Let initial output is O then after 10% increase it will be 1.1O and after 20% increase on this new output the latest output will be 1.1O * 1.2O = 1.32O. Now we have to decrease the output by some percentage so that the new output is same as the starting output (O). so, 1.32O * (1-x/100) = O. => x = 24.24%. So, answer will be B. The answer is (b). 

 In a graduate physics course, 70 percent of the students are male and 30 percent of the students are married. If two-sevenths of the male students are married, what fraction of the male students is single? Answer Choices: (a) 2/7 (b) 1/3 (c) 1/2 (d) 2/3 (e) 5/7 
 A: let assume there are 100 students of which 70 are male and 30 are females. if 30 are married then 70 will be single. now its given that two-sevenths of the male students are married that means 2/7 of 70 = 20 males are married. if 30 is the total number of students who are married and out of that 20 are males then the remaining 10 will be females who are married. total females = 70. married males = 20. then single males = 70-20 = 50. we need to find the fraction of male students who are single i.e single male students / total male student = 50/70 = 5/7 [E]. The answer is (e). |

Table 11: One example of selected model-generated exemplars with rationale chains. Note that there are newlines between the answer choices which are omitted in the table to save space.

| DATASET | Exemplars |
|---|---|
| CSQA | Q: John was punching and punching at the wall but succeeded only in bloodying his knuckles. This was bad. He would be unable to hold his tools if he injured what? Answer Choices: (a) hostility (b) anger (c) nose (d) fists (e) hands
A: The answer should be something that John would be unable to hold his tools if he injured it. Of the above choices, only hands make sense. So the answer is (e).

Q: What is required to be good at playing tennis? Answer Choices: (a) tennis racket (b) skill (c) competition (d) tennis court (e) legs
A: The answer must be something that is required to be good at playing tennis. Of the above choices, only skill is required to be good at playing tennis. So the answer is (b).

Q: What is someone who gets angry after getting drunk likely to participate in? Answer Choices: (a) violence (b) falling down (c) vomiting (d) vomiting (e) lower standards
A: The answer should be something that someone who gets angry after getting drunk is likely to do. Of the above choices, the closest answer is violence. So the answer is (a).

Q: James felt himself losing consciousness. He was frightened. He didn't want this. He was scared of what? Answer Choices: (a) death (b) coma (c) peace (d) forgetfulness (e) loss of life
A: The answer should be something that James would be scared of. Of the above choices, the closest thing is death. So the answer is (a).

Q: Sally lost her kite because she wasn't careful. She thought that there was more string on the spool, but it slipped out of her fingers when she reached the what?. Answer Choices: (a) child's hand (b) the last straw (c) hobby shop (d) end of line (e) toy store
A: The answer should be the end of the string. Of the above choices, the end of the line makes the most sense. So the answer is (d).

Q: The sensor would just the distance then set off an alarm, the installation expert explained it was called a what kind of sensor? Answer Choices: (a) near (b) closeness (c) here (d) proximity (e) this
A: The answer must be a type of sensor that would set off an alarm when something is near. Of the above choices, only proximity sensor makes sense. So the answer is (d).

Q: The student needed to get some new pencils, where did he go? Answer Choices: (a) store (b) classroom (c) backpack (d) cabinet (e) teacher
A: The answer should be the place where pencils are sold. Of the above choices, only store makes sense. So the answer is (a). |

Table 12: One example of selected model-generated exemplars with rationale chains. This set of exemplars is trained and selected on CommonsenseQA. Note that there are newlines between the answer choices which are omitted in the table to save space.

| DATASET | Exemplars |
|---------|-----------|
| StrategyQA | Q: Did the death of Helen Palmer have a significant effect on Dr. Seuss?
A: Helen Palmer was Dr. Seuss's wife. The death of a spouse can have a significant effect on a person. Thus, the death of Helen Palmer could have a significant effect on Dr. Seuss. So the answer is yes.

Q: Would a moose hypothetically be too much for a minotaur to devour whole?
A: A moose is about 1,000 pounds. A minotaur is about 1,000 pounds. Thus, a moose would be too much for a minotaur to devour whole. So the answer is yes.

Q: Yes or no: Is H's most common two letter pair partner a freebie in Wheel of Fortune bonus round?
A: The most common two letter pair partner for H is E. E is a freebie in the Wheel of Fortune bonus round. Thus, H's most common two letter pair partner is a freebie in the Wheel of Fortune bonus round. So the answer is yes.

Q: Yes or no: Would bald eagle deliver an urgent message before B-52?
A: The top speed of a bald eagle is about 80 mph. The top speed of a B-52 is about 650 mph. Thus, a B-52 would be faster than a bald eagle. So the answer is no.

Q: Yes or no: Does an individual oceanographer study many sciences?
A: Oceanography is the study of the ocean. The ocean is a complex system. Thus, oceanography is a multidisciplinary science. So the answer is yes.

Q: Yes or no: Could Steven Spielberg send emails as a child?
A: Steven Spielberg was born in 1946. Email was invented in 1971. Thus, Steven Spielberg could not send emails as a child. So the answer is no. |

Table 13: One example of selected model-generated exemplars with rationale chains. This set of exemplars is trained and selected on StrategyQA. Note that there are newlines between the answer choices which are omitted in the table to save space.

| DATASET | Exemplars |
|---------|-----------|
| Letter (4) | Q: Take the last letters of the words in "Delmer Andersen" and concatenate them.
A: The last letter of "Delmer" is "r". The last letter of "Andersen" is "n". Concatenating them is "rn". So the answer is rn.

Q: Take the last letters of the words in "Jason Gonzalez" and concatenate them.
A: The last letter of "Jason" is "n". The last letter of "Gonzalez" is "z". Concatenating them is "nz". So the answer is nz.

Q: Take the last letters of the words in "Ulysses Brown" and concatenate them.
A: The last letter of "Ulysses" is "s". The last letter of "Brown" is "n". Concatenating them is "sn". So the answer is sn.

Q: Take the last letters of the words in "Frank Ortiz" and concatenate them.
A: The last letter of "Frank" is "k". The last letter of "Ortiz" is "z". Concatenating them is "kz". So the answer is kz. |

Table 14: One example of selected model-generated exemplars with rationale chains. This set of exemplars is trained on Letter (3) and selected on Letter (2).

| DATASET | Exemplars |
|---|---|
| OpenBookQA | Q: As a car approaches you in the night (a) the headlights become more intense (b) the headlights recede into the dark (c) the headlights remain at a constant (d) the headlights turn off
A: The headlights become more intense as the car approaches. The answer is (a).

Q: Many animals that give birth to live young have (a) gills (b) scales (c) exoskeletons (d) legs
A: Animals that give birth to live young are mammals. Mammals have hair and give milk to their young. The answer is (a).

Q: A person is lost in a dense forest, and needs to find their home. They know their home is to the south, and they are headed north. They can find home by using a (a) northern-directing device (b) northern light reader (c) northeastern winds (d) north central credit
A: he person needs to find south, so they need a northern-directing device. The answer is (a).

Q: When the weather changes as it does from Christmas to Easter, (a) the air may chill (b) the ground may freeze (c) the plants may die (d) the ground may warm
A: The weather changes from Christmas to Easter, the ground may warm. The answer is (d). |

Table 15: One example of selected exemplars with rationale chains. This set of exemplars is trained and selected on OpenBookQA.

| DATASET | Exemplars |
|---------|-----------|
| e-SNLI | Premise:
"A woman wearing all white and eating, walks next to a man holding a briefcase."
Based on this premise,can we conclude the hypothesis "A female is next to a man." is true?
OPTIONS:
- yes
- no
- it is not possible to tell
A: A woman is another word for a female. The answer is yes.

Premise:
"A couple play in the tide with their young son."
Based on this premise,can we conclude the hypothesis "The family is sitting down for dinner." is true?
OPTIONS:
- yes
- no
- it is not possible to tell
A: The family cannot simultaneously be playing in the tide and sitting down to dinner The answer is no.

Premise:
"A man, woman, and child enjoying themselves on a beach."
Based on this premise,can we conclude the hypothesis "A child with mom and dad, on summer vacation at the beach." is true?
OPTIONS:
- yes
- no
- it is not possible to tell
A: A man and a woman with a child are not necessarily its mom and dad. A man, woman, and child at a beach are not necessarily on summer vacation. The answer is it is not possible to tell.

Premise:
"Two women, holding food carryout containers, hug."
Based on this premise,can we conclude the hypothesis "The two sisters saw each other across the crowded diner and shared a hug, both clutching their doggie bags." is true?
OPTIONS:
- yes
- no
- it is not possible to tell
A: Just because two women are hugging does not mean they are sisters or they are in a crowded diner. The answer is it is not possible to tell.

Premise:
"A few people in a restaurant setting, one of them is drinking orange juice."
Based on this premise,can we conclude the hypothesis "The people are eating omelettes." is true?
OPTIONS:
- yes
- no
- it is not possible to tell
A: We only know that someone is drinking orange juice, we do not know that they are eating omelettes The answer is it is not possible to tell.

Premise:
"A man and a woman cross the street in front of a pizza and gyro restaurant."
Based on this premise,can we conclude the hypothesis "Near a couple of restaurants, two people walk across the street." is true?
OPTIONS:
- yes
- no
- it is not possible to tell
A: man and woman are people. The answer is yes. |

Table 16: One example of selected exemplars with rationale chains. This set of exemplars is trained and selected on e-SNLI.

| DATASET | Exemplars |
|---|---|
| SST-2 | What is the sentiment of the following sentence?
"more than another " best man " clone by weaving a theme throughout this funny film"
A: "weaving a theme throughout this funny film" indicates positive sentiment. The answer is positive.

What is the sentiment of the following sentence?
"that 's far too tragic to merit such superficial treatment"
A: "far too tragic" and "to merit such superficial treatment" both mean negative sentiments. The answer is negative.

What is the sentiment of the following sentence?
"are more deeply thought through than in most ' right-thinking ' films"
A: "more deeply thought through" indicates positive sentiment. The answer is positive.

What is the sentiment of the following sentence?
"excruciatingly unfunny and pitifully unromantic"
A: "excruciatingly unfunny" and "pitifully unromantic" both mean negative sentiments. The answer is negative..

What is the sentiment of the following sentence?
"with his usual intelligence and subtlety"
A: "with his usual intelligence and subtlety" indicates positive sentiment. The answer is positive.

What is the sentiment of the following sentence?
"goes to absurd lengths"
A: "goes to absurd lengths" is a negative sentiment. The answer is negative. |

Table 17: One example of selected exemplars with rationale chains. This set of exemplars is trained and selected on SST-2.