# OpenReview forum: "Automatic Prompt Augmentation and Selection with Chain-of-Thought from Labeled Data"
_EMNLP/2023/Conference — EMNLP 2023 Findings_

### Official Review · Reviewer_5T7Y · 2023-07-20

**Soundness:** 4

**Excitement:**

4: Strong: This paper deepens the understanding of some phenomenon or lowers the barriers to an existing research direction.

**Paper Topic And Main Contributions:**

This work aims to address the restriction of handcrafting effective input-output demonstrations in most chain-of-thought (CoT) studies, where the effectiveness of demonstrations may be influenced by four aspects: order sensitivity, complexity, diversity, and style sensitivity. To this end, this paper presents an Automate-CoT strategy to bypass human engineering of CoT by automatically augmenting rational chains from a small labeled dataset, and then pruning low-quality chains to construct a candidate pool of machine-generated rationale chains based on the labels. The proposed approach works by three steps, augment, prune, and select. It first generates rationale chains according to the standard CoT process with several exemplars, and then prunes the incorrect ones according to the consistency of the predicted answer and ground-truth answer. Finally, a variance-reduced policy gradient strategy is applied to estimate the gradients and optimize the latent variables to select better CoTs.

Overall, this work may advance the development of automatic and task-adaptive prompting CoT methods that can be generally effective across large language models. Though sharing a similar motivation, this work makes solid contributions over the existing Auto-CoT approach, with improvement gains over a range of tasks. This work also provides insightful analysis that justifies the contribution factors and the (in)sensitivity of the proposed approach.

**Questions For The Authors:**

Manual-CoT, Zero-Shot-CoT and Auto-CoT all find them performing inferiorly in CommonsenseQA compared with direct reasoning. In contrast, this work achieves remarkable performance gains in that dataset. Is there any evidence to justify the advantage in this aspect?

**Reasons To Accept:**

1. This paper is generally written in a clear and concise style. This work is well motivated, and provides a useful alternative of automatic CoT prompting approach. The relationship with existing studies is also clearly presented.

2. The experimental results are impressive. The proposed approach achieves satisfying performance gains over the baselines under a relatively fair comparison. Analysis shows that the approach works robustly.

**Reasons To Reject:**

1. Some claims may be inspired from existing studies; thus, it is critical to add the supportive references. For example, Lines 55-64: "we identify four critical factors that affect the performance of chain-of-thought prompting and require large human effort to deal with: (1) order sensitivity: the order combination of the exemplars; (2) complexity: the number of reasoning steps of the rationale chains; (3) diversity: the combination of different complex-level exemplars; (4) style sensitivity: the writing/linguistic style of the rationale chains." --- Most of the above factors have been discussed in existing studies.

2. This approach requires extensive queries to optimize and organize the demonstration exemplars, which would costly behind the paywalls. It also relies on a training-based pipeline, which further increases the complexity of the whole framework.

**Reproducibility:**

4: Could mostly reproduce the results, but there may be some variation because of sample variance or minor variations in their interpretation of the protocol or method.

**Reviewer Confidence:**

4: Quite sure. I tried to check the important points carefully. It's unlikely, though conceivable, that I missed something that should affect my ratings.

---

> ### Author Rebuttal · Authors · 2023-08-29
>
> Dear Reviewer 5T7Y,
>
> Thank you very much for your comprehensive review and valuable feedback! We address your comments one by one as follows:
>
> **[Add supportive references]**
> > Some claims may be inspired from existing studies; thus, it is critical to add the supportive references. Most of the above factors have been discussed in existing studies.
>
> Thanks for your suggestions! Most of the factors have been discussed in existing studies and we have updated this paragraph to:
> "we identify four critical factors that affect the performance of chain-of-thought prompting and require large human effort to deal with: (1) order sensitivity [1]: the order combination of the exemplars; (2) complexity [2-4]: the number of reasoning steps of the rationale chains; (3) diversity: the combination of different complex-level exemplars; (4) style sensitivity [1,5]: the writing/linguistic style of the rationale chains."
> If we are missing any literature, please let us know. We are happy to discuss and cite them. Thank you!
>
> [1] Zhao, Z., Wallace, E., Feng, S., Klein, D., & Singh, S. (2021, July). Calibrate before use: Improving few-shot performance of language models. In International Conference on Machine Learning (pp. 12697-12706). PMLR.
>
> [2] Fu, Y., Peng, H., Sabharwal, A., Clark, P., & Khot, T. (2022). Complexity-based prompting for multi-step reasoning. arXiv preprint arXiv:2210.00720.
>
> [3] Sugawara, S., Inui, K., Sekine, S., & Aizawa, A. (2018). What makes reading comprehension questions easier?. arXiv preprint arXiv:1808.09384.
>
> [4] Lai, Y., Zhang, C., Feng, Y., Huang, Q., & Zhao, D. (2021). Why machine reading comprehension models learn shortcuts?. arXiv preprint arXiv:2106.01024.?
>
> [5] Papadopoulos, P. M., Demetriadis, S. N., Stamelos, I. G., & Tsoukalas, I. A. (2010). The effect of prompting to students with different learning styles. Multicultural Education & Technology Journal, 4(3), 198-213.
>
>
>
> **[Cost]**
>
> > This approach requires extensive queries to optimize and organize the demonstration exemplars, which would costly behind the paywalls. It also relies on a training-based pipeline, which further increases the complexity of the whole framework.
>
>
> Thanks for your suggestion! We provide the estimated cost of optimizing the prompts in terms of money with gpt-3.5-turbo. The usage of gpt-3.5-turbo is 0.0015 USD / 1K tokens for input and 0.002 USD / 1K tokens for output tokens. Under Automate-CoT with the training epochs of 3, a training set size of 100 and a validation set size of 100, an input length of around 750 tokens and an average output length of 150 tokens, it takes about (750/1000 * 0.0015 + 150/1000 * 0.002) * 100 * 10 * 3  + (750/1000 * 0.0015 + 150/1000 * 0.002) * 100 * 3 = 4.7 USD.
>
>
> We have updated our manuscript with these details. Thanks very much for your constructive suggestion!
>
>
> **[Performance on CommonsenseQA]**
>
> > Manual-CoT, Zero-Shot-CoT and Auto-CoT all find them performing inferiorly in CommonsenseQA compared with direct reasoning. In contrast, this work achieves remarkable performance gains in that dataset. Is there any evidence to justify the advantage in this aspect?
>
> There are two potential reasons:
> On the CSQA dataset, the quality of the original manual constructed CoT is mediocre. As shown in our Figure 3, we found that even  the Random Selection method can performs better on CSQA than Manual-CoT many times.
> And because our proposed selection can help to choose appropriate few-shot examples, addressing the issues of order and diversity, the performance has been significantly improved.
>
> We have updated our manuscript with these new analyses and discussions. Thanks very much for your kind and constructive suggestions!

---

### Official Review · Reviewer_rCCa · 2023-08-02

**Typos Grammar Style And Presentation Improvements:** NA
**Soundness:** 4

**Excitement:**

3: Ambivalent: It has merits (e.g., it reports state-of-the-art results, the idea is nice), but there are key weaknesses (e.g., it describes incremental work), and it can significantly benefit from another round of revision. However, I won't object to accepting it if my co-reviewers champion it.

**Missing References:**

[1] Learning To Retrieve Prompts for In-Context Learning. EMNLP 2023.

[2] Selective annotation makes language models better few-shot learners. ICLR 2023.

[3] Dynamic Prompt Learning via Policy Gradient for Semi-structured Mathematical Reasoning. ICLR 2023.

**Paper Topic And Main Contributions:**

The paper introduces a new strategy called Automate-CoT to enhance the reasoning abilities of large language models (LLMs). Automate-CoT addresses this manual limitation by automatically augmenting rational chains from a small training dataset, pruning low-quality chains, and selecting the optimal combination of rationale chains using a variance-reduced policy gradient strategy. The approach is evaluated on various reasoning tasks and non-reasoning tasks, demonstrating competitive performance improvements.

**Questions For The Authors:**

1. Line 240&241. What are T, t_i, S for? The loss is poorly explained.

2. How does the selection method aid in identifying diverse and representative samples?

3. Recent studies on in-context learning have shown that dynamic demonstration example selection strategies may be superior to fixed demonstration example selection strategies [1]. Auto-CoT compares not only random strategies but also retrieval strategies. It would be beneficial to compare the mentioned selection with simple retrieval methods like BM25 or Learn to Retrieve [1], Vote-K [2], RL [3].

5. Auto-CoT selects demonstration examples through clustering. Have this paper considered comparing the selection method to clustering?

6. In Figure 4, it is evident that performance improves with an increase in pool size. However, it is important to determine the specific pool size at which performance convergence occurs or starts to drop. The cost of running 200 queries on GPT-3 is acceptable.

7. The experiment datasets chosen in the additional experiment section of the paper are not consistent. Specifically, in Table 2, the paper opts for GSM8K, SVAMP, and Letter, in Table 3, it selects GSM8K, CSQA, and Letter, and in Table 4, it again chooses GSM8K, SVAMP, and Letter.

8. In table 3, Auto-CoT should be a more relevant baseline since Auto-CoT is a zero-shot baseline without using ground-truth labels.

I will update the final score according to the authors responses.

**Reasons To Accept:**

1. The research topic regarding example selection for CoT is important, and the motivation behind this paper is clear.

2. The idea is new and practical for CoT research.

3. Extensive experimental results and ablation studies demonstrate the effectiveness of the proposed method.

**Reasons To Reject:**

1. The paper missed some strong prompt selection baselines for few-shot learning.

2. The notations in section 3.2 of the paper are poorly defined and explained, making it difficult to read.

3. As mentioned in the Limitations section, the paper does not include experiments with recent closed-source LLMs, such as ChatGPT. It is worth noting that ChatGPT may be both cheaper and stronger than text-davinci-002.

4. The relationship between the proposed method and the four motivation factors is still unclear.

**Reproducibility:**

3: Could reproduce the results with some difficulty. The settings of parameters are underspecified or subjectively determined; the training/evaluation data are not widely available.

**Reviewer Confidence:**

4: Quite sure. I tried to check the important points carefully. It's unlikely, though conceivable, that I missed something that should affect my ratings.

---

> ### Author Rebuttal · Authors · 2023-08-29
>
> Dear Reviewer rCCa,
>
> Thank you very much for your comprehensive review and valuable feedback! We address your comments one by one as follows:
>
>
> **[Comparison with retrieval-based method]**
>
> > The paper missed some strong prompt selection baselines for few-shot learning. Recent studies on in-context learning have shown that dynamic demonstration example selection strategies may be superior to fixed demonstration example selection strategies [1]. Auto-CoT compares not only random strategies but also retrieval strategies. It would be beneficial to compare the mentioned selection with simple retrieval methods like BM25 or Learn to Retrieve [1], Vote-K [2], RL [3].
>
> Thanks for your suggestion! We recognize adding the comparison with the retrieval-based selection method would enhance the persuasiveness of our research. Following your advice, we implemented a BM25-based selection method and tested the performance of all the datasets. The results are shown below. It indicates that retrieval-based methods can only select examples with similar meaning to the query question while diversity is overlooked. As shown in the table, the average performance of the BM25 retrieval-based method even has a one percent degradation compared to Manual-CoT, and 3.8% lower than Automate-CoT. A similar phenomenon is observed in Auto-CoT, which indicates that with similar questions being sampled for test questions, Retrieval-Q-CoT is negatively affected by misleading by similarity.
>
> | | GSM8K | ASDiv | SVAMP | AQuA | SingleOP | CSQA | StrategyQA | Letter (4) | OBQA | e-SNLI | SST-2 | Avg. |
> | ----------- | ----------- | ----------- | ----------- | ----------- | ----------- | ----------- | ----------- | ----------- | ----------- | ----------- | ----------- | ----------- |
> | Manual-CoT  | 63.1 | 77.1 | 78.1 | 44.9 | 90.0 | 77.5 | 59.7 | 73.0 | 80.0 | 80.9 | 85.3 | 73.6 |
> | BM25  | 64.2| 73.7 | 73.8 | 45.3 | 87.9 | 76.1 | 58.9 | 73.4 | 81.4 | 76.3 | 87.2 | 72.6 |
> | Automate-CoT | 68.0 | 81.7 | 79.1 | 46.9 | 91.5 | 80.5 | 64.5 | 76.2 | 83.0 | 81.4 | 87.7 | 76.4 |
>
> We have updated our manuscript with these new results, to reveal the effects of retrieval-based methods. Thanks very much for your constructive suggestion!
>
>
> **[Notations in section 3.2]**
>
> > Line 240&241. What are T, t_i, S for? The loss is poorly explained.
>
> In Lines 240&241, T are the full few-shot exemplars. t_i denotes the $i$-th exemplar. S is the current question (user’s query). We first sample each exemplar t_i according to P(t_i). After obtaining T, we concatenate T with the query input S and feed [T, S] into the large language model G to get a prediction. Then we compare the prediction with $y$ and calculate the loss. Thank you for pointing it out! We have updated our manuscript with these details.
>
> **[Experiments with gpt-3.5-turbo]**
>
>
> > As mentioned in the Limitations section, the paper does not include experiments with recent closed-source LLMs, such as ChatGPT. It is worth noting that ChatGPT may be both cheaper and stronger than text-davinci-002.
>
>
> We further conducted the experiments on gpt-3.5-turbo. Our Automate-CoT also shows consistent improvement on each task with 2.8% improvement on arithmetic reasoning, 3,9% improvement on commonsense reasoning, 3.2% on symbolic reasoning, and 2.8% improvement overall.
>
> | | GSM8K | ASDiv | SVAMP | AQuA | SingleOP | CSQA | StrategyQA | Letter (4) | OBQA | e-SNLI | SST-2 | Avg. |
> | ----------- | ----------- | ----------- | ----------- | ----------- | ----------- | ----------- | ----------- | ----------- | ----------- | ----------- | ----------- | ----------- |
> | Manual-CoT  | 63.1 | 77.1 | 78.1 | 44.9 | 90.0 | 77.5 | 59.7 | 73.0 | 80.0 | 80.9 | 85.3 | 73.6 |
> | Automate-CoT | 68.0 (+4.9) | 81.7 (+4.6) | 79.1 (+1.0) | 46.9 (+2.0) | 91.5 (+1.5) | 80.5 (+3.0) | 64.5 (+4.8) | 76.2 (+3.2) | 83.0 (+3.0) | 81.4 (+0.5) | 87.7 (+2.4) | 76.4 (+2.8) |
>
> We have updated our manuscript with these new results. Thanks very much for your constructive suggestion!
>
>
> **[Four motivation factors]**
>
> > The relationship between the proposed method and the four motivation factors is still unclear. How does the selection method aid in identifying diverse and representative samples?
>
> In Section 2, we first investigate the existence of the sensitivities in chain-of-thought methods. Then we further explore other factors that would not only affect the performance but require human efforts to deal with. In light of this empirical evidence, we are motivated to design a framework not only to augment the rationale chains but also to select the helpful rationale chains adaptively.
> In Section 7, we analyze the effects of these four factors with our proposed method. We found that Automate-CoT can bypass the order and style sensitivity issues and reach a better complexity-diversity trade-off without human effort, finally boosting performance.
>
> Especially, in Section 2, we found that diverse and representative samples are helpful in math reasoning tasks. During our model’s selection process, the exemplars are optimized and it will automatically learn to select the most helpful exemplars.
>
> Thank you for your question and hope this explanation addresses your comments!
>
>
>
> **[Comparison with clustering method]**
>
> > Auto-CoT selects demonstration examples through clustering. Have this paper considered comparing the selection method to clustering?
> Thanks for your suggestion! We recognize adding the comparison with the clustering-based selection method would enhance the persuasiveness of our research. Following your advice, we implemented a K-means-based selection method and tested the performance of all the datasets. We first use OpenAI’s `text-embedding-ada-002` embedding API to compute a vector representation for each question in the pool and then we use K-means to produce k clusters of questions. Finally, we select one representative question for each cluster by choosing the question with minimal distance to the cluster center.
>
> The results are shown below. It indicates that clustering-based methods can select examples with different semantic meanings and generally perform better than Manual-CoT. However, the complexity and diversity are overlooked. For example, most of the selected few-shot exemplars in GSM8K have around 3-4 hops where complex questions and moderately difficult questions are overlooked. As a result, it generally performs worse than Automate-CoT with a 2.6% gap. We have updated our manuscript with these new results, to reveal the effects of clustering-based methods. Thanks very much for your constructive suggestion!
>
> | | GSM8K | ASDiv | SVAMP | AQuA | SingleOP | CSQA | StrategyQA | Letter (4) | OBQA | e-SNLI | SST-2 | Avg. |
> | ----------- | ----------- | ----------- | ----------- | ----------- | ----------- | ----------- | ----------- | ----------- | ----------- | ----------- | ----------- | ----------- |
> | Manual-CoT  | 63.1 | 77.1 | 78.1 | 44.9 | 90.0 | 77.5 | 59.7 | 73.0 | 80.0 | 80.9 | 85.3 | 73.6 |
> | K-Means | 66.4 | 76.6 | 77.6 | 45.7 | 89.7 | 79.0 | 60.0 | 73.6 | 80.4 | 78.4 | 84.1 | 73.8 |
> | Automate-CoT | 68.0 (+4.9) | 81.7 (+4.6) | 79.1 (+1.0) | 46.9 (+2.0) | 91.5 (+1.5) | 80.5 (+3.0) | 64.5 (+4.8) | 76.2 (+3.2) | 83.0 (+3.0) | 81.4 (+0.5) | 87.7 (+2.4) | 76.4 (+2.8) |
>
>
>
>
> **[Additional experiments on pool size]**
>
> > In Figure 4, it is evident that performance improves with an increase in pool size. However, it is important to determine the specific pool size at which performance convergence occurs or starts to drop. The cost of running 200 queries on GPT-3 is acceptable.
>
> Previously, our experiments on pool size were conducted under code-davinci-002 to save cost. However, OpenAI has decided to shut off access to code-davinci-002 in March 2023. Therefore, we further re-conducted the same setting on gpt-3.5-turbo and added an experiment with a pool size of 200 into comparison.
> | Datasets\ Pool Size  | 10 | 20 | 50 | 100 | 150 | 200|
> | ----------- | ----------- | ----------- | ----------- | ----------- | ----------- | ----------- |
> | GSM8K | 63.0 | 65.4 | 66.7 | 68.0 | 68.4 | 68.3 |
> | SVAMP | 76.0 | 76.9 | 78.0 | 79.1 | 79.5 | 79.6 |
> | Letter(4) | 69.6 | 71.2 | 73.8 | 76.2 | 76.8 | 76.4 |
> According to the result, it shows that when the pool size approaches 200, the performance on three datasets shows minimal increase or even a little bit drop. This shows that a pool size of 100-150 is enough for the model to find a good combination for the task.
> We have updated our manuscript with these new results. Thanks very much for your constructive suggestion!
>
>
> **[Change of datasets in the additional experiments]**
>
> > The experiment datasets chosen in the additional experiment section of the paper are not consistent. Specifically, in Table 2, the paper opts for GSM8K, SVAMP, and Letter, in Table 3, it selects GSM8K, CSQA, and Letter, and in Table 4, it again chooses GSM8K, SVAMP, and Letter.
>
> Thank you for your reminder of the inconsistent datasets chosen in the additional experiment. We now change the dataset in the additional zero-shot experiment (Table 3) to GSM8K, SVAMP and Letter(4) to be consistent with Table 2 and Table 4.
> The results are in the table below:
>
> | | GSM8K | SVAMP | Letter(4) |Avg.|
> | ----------- | ----------- | ----------- | ----------- |  ----------- |
> | Zero-Shot-CoT| 40.7 | 62.1 | 57.6 | 53.5 |
> | Manual-CoT | 46.9 | 73.5 | 56.6 | 59 |
> | Auto-CoT | 48.9 | 69.5 | 59.7 | 59.4 |
> | Zero-Shot-Automate-CoT | 49.1 | 72.4 | 59.3 | 60.3 |
>
>
> **[Additional baseline comparison in Table 3]**
>
> > In table 3, Auto-CoT should be a more relevant baseline since Auto-CoT is a zero-shot baseline without using ground-truth labels.
>
> After changing the dataset following the previous question, we also added Auto-CoT into this comparison.
> | | GSM8K | SVAMP | Letter(4) | Avg.|
> | ----------- | ----------- | ----------- | ----------- | ----------- |
> | Zero-Shot-CoT| 40.7 | 62.1 | 57.6 |53.5|
> | Manual-CoT | 46.9 | 73.5 | 56.6 |59|
> | Auto-CoT| 48.9 | 69.5 | 59.7 | 59.4 |
> | Zero-Shot-Automate-CoT| 49.1 | 72.4 | 59.3 | 60.3 |
>
> The results show that Zero-shot-Automate-CoT has an advantage of 0.9% on average than Auto-CoT. In addition to text-davinci-002, since Auto-CoT only reports results of gsm8k under code-davinci-002, we also report zero-shot Automate-CoT of gsm8k under code-davinci-002 for comparison.
>
> | | GSM8K |
> | ----------- | ----------- |
> | | code-davinci-002 |
> | Auto-CoT | 62.8 |
> | Zero-Shot-Automate-CoT | 65.8 |
>
> We have updated our manuscript with these new results. Thanks very much for your constructive suggestion!

---

### Official Review · Reviewer_1kzZ · 2023-08-05

**Soundness:** 4

**Excitement:**

3: Ambivalent: It has merits (e.g., it reports state-of-the-art results, the idea is nice), but there are key weaknesses (e.g., it describes incremental work), and it can significantly benefit from another round of revision. However, I won't object to accepting it if my co-reviewers champion it.

**Paper Topic And Main Contributions:**

This paper studies the problem of automatically engineering chain-of-thoughts prompts using labeled data. The paper proposes Automate-CoT, which takes as input a set of labeled data points and, optionally, a few data points with annotated chain-of-thoughts, and automatically outputs an optimized CoT prompts. The paper tests the proposed approach on 11 datasets and compares against AutoCoT and manual CoT baseline. The results suggest the proposed Automate-CoT approach can effectively leverage labeled data points to optimize the CoT prompts.

**Questions For The Authors:**

Question A: Can you elaborate on the cost of optimizing prompts?

**Reasons To Accept:**

The paper provides a way to automate the engineering of CoT prompts and achieves noticeable improvements over manually crafted CoT prompts.

The experiments cover multiple datasets and domains and also provide thorough analysis and ablations.

The paper is well-written and easy to follow.

**Reasons To Reject:**

While the proposed approach leads to some improvements over manually crafted CoT prompts, it requires 1) hundreds of labeled data points to work well 2) non-negligible computation overheads for optimizing the prompts. One advantage of CoT prompting is its applicability in true few-shot settings; requiring hundreds of labeled data points could be practical in some scenarios but also limits the applicability of the proposed approach to some extent. Regarding (2), it would be good if the paper can elaborate on the cost of optimizing prompts measured by money or number of tokens beyond specifying hyper-parameters.

The proposed approach is a straightforward incremental extension of AutoCoT (Zhang et al., 2022). In particular, the augment and prune are the same as in AutoCoT (Zhang et al., 2022) and Rationale-Augmented Ensemble (Wang et al., 2022). The only added component is a derivative-free optimizer that can leverage labeled data for selecting examples at some additional computational cost.

While it has been mentioned in the limitation, it could be helpful if the paper can experiment with more advanced models like gpt-3.5-turbo.

**Reproducibility:**

3: Could reproduce the results with some difficulty. The settings of parameters are underspecified or subjectively determined; the training/evaluation data are not widely available.

**Reviewer Confidence:**

4: Quite sure. I tried to check the important points carefully. It's unlikely, though conceivable, that I missed something that should affect my ratings.

---

> ### Author Rebuttal · Authors · 2023-08-29
>
> Dear Reviewer 1kzZ,
>
> Thank you very much for your comprehensive review and valuable feedback! We address your comments one by one as follows:
>
> **[Requirement of labeled data]**
> > While the proposed approach leads to some improvements over manually crafted CoT prompts, it requires 1) hundreds of labeled data points to work well. One advantage of CoT prompting is its applicability in true few-shot settings; requiring hundreds of labeled data points could be practical in some scenarios but also limits the applicability of the proposed approach to some extent.
>
>
> Thanks for your question! Our method only needs question-answer pairs instead of rational chains. In the real world, there are many scenarios and applications where labeled question-answer pairs are available without human-annotated rational chains. Considering that this method requires only a small amount of labeled data to achieve good results, especially in alleviating the sensitivities associated with the traditional Chain-of-Thought approach, it is worthwhile to apply Automate-CoT.
>
>
>
>
> **[Cost of optimizing prompts]**
>
>
> > 2) non-negligible computation overheads for optimizing the prompts. Regarding (2), it would be good if the paper can elaborate on the cost of optimizing prompts measured by money or number of tokens beyond specifying hyper-parameters.
>
>
> Thanks for your suggestion! We provide the estimated cost of optimizing the prompts in terms of money with gpt-3.5-turbo. The usage of gpt-3.5-turbo is 0.0015 USD / 1K tokens for input and 0.002 USD / 1K tokens for output tokens. Under Automate-CoT with the training epochs of 3, a training set size of 100 and a validation set size of 100, an input length of around 750 tokens and an average output length of 150 tokens, it takes about (750/1000 * 0.0015 + 150/1000 * 0.002) * 100 * 10 * 3  + (750/1000 * 0.0015 + 150/1000 * 0.002) * 100 * 3 = $4.7.
>
>
> We have updated our manuscript with these details. Thanks very much for your constructive suggestion!
>
>
>
>
> **[Contribution]**
>
>
> > The proposed approach is a straightforward incremental extension of AutoCoT (Zhang et al., 2022). In particular, the augment and prune are the same as in AutoCoT (Zhang et al., 2022) and Rationale-Augmented Ensemble (Wang et al., 2022). The only added component is a derivative-free optimizer that can leverage labeled data for selecting examples at some additional computational cost.
>
>
> Our proposed method enhances the reasoning abilities by automatically augmenting rational chains from a small training dataset, pruning low-quality chains, and selecting the optimal combination of rationale chains using a variance-reduced policy gradient strategy.
> The augmentation and pruning methods that we are using are different from both Auto-CoT and Rationale-Augmented Ensemble.
>
>
> Our method differs from Auto-CoT in several aspects:
> [Augment] Auto-CoT mainly augments the questions by leveraging “Let’s think step by step” and further it directly uses the generated rationales as the model input that contains wrong and low-quality chains. However, Automate-CoT augments the results by using two different methods, the few-shot and zero-shot method.
>
> [Prune] Auto-CoT does not have the pruning step while we propose to prune the low-quality chains to construct a high-quality candidate pool.
>
> [Select] More importantly, we propose a new method for selecting the optimal combination of rationale chains using a variance-reduced policy gradient strategy.
>
> [Setting] For the experimental settings, Auto-CoT directly samples questions from the test dataset while we restrict our method to training data which is more realistic in applications (considering most test set is hidden before testing).
>
>
> Our method differs from Rationale-Augmented Ensemble in several aspects:
> They have shown that randomly replacing one human-written out of K (4-6) exemplars exhibits substantial variance which means human-written rationales can be far from “optimal”. As a result, their method focuses on “ensemble” which elicits multiple rationales (and outputs) by using such replace method and ensemble the results. For each sampled path, the few-shot exemplars are still far away from “optimal”, and even worse than manually constructed ones( as mentioned in the Rational-augmented Ensemble paper Section 2.1). However, our method focuses on getting the optimal few-shot exemplars where using a selection-based method to mitigate the low-quality chains and further bypass the four sensitivities.
>
>
> **[Experiments with gpt-3.5-turbo]**
>
>
> > While it has been mentioned in the limitation, it could be helpful if the paper can experiment with more advanced models like gpt-3.5-turbo.
>
>
> We further conduct the experiments on gpt-3.5-turbo, our Automate-CoT also shows consistent improvement on each task with 2.8% improvement on arithmetic reasoning, 3.9% improvement on commonsense reasoning, 3.2% on symbolic reasoning and 2.8% improvement overall.
>
> | | GSM8K | ASDiv | SVAMP | AQuA | SingleOP | CSQA | StrategyQA | Letter (4) | OBQA | e-SNLI | SST-2 | Avg. |
> | ----------- | ----------- | ----------- | ----------- | ----------- | ----------- | ----------- | ----------- | ----------- | ----------- | ----------- | ----------- | ----------- |
> | Manual-CoT  | 63.1 | 77.1 | 78.1 | 44.9 | 90.0 | 77.5 | 59.7 | 73.0 | 80.0 | 80.9 | 85.3 | 73.6 |
> | +Automate-CoT | 68.0 (+4.9) | 81.7 (+4.6) | 79.1 (+1.0) | 46.9 (+2.0) | 91.5 (+1.5) | 80.5 (+3.0) | 64.5 (+4.8) | 76.2 (+3.2) | 83.0 (+3.0) | 81.4 (+0.5) | 87.7 (+2.4) | 76.4 (+2.8) |
>
> We have updated our manuscript with these new results. Thanks very much for your constructive suggestion!

---

### Meta-Review · Area_Chair_CGKk · 2023-09-18

**Recommendation:** 3

**Metareview:**

This paper presented a work with a few nice contributions as mentioned by the reviewers and summarized by the authors during discussions, which I would prefer not to repeat. I do have a few additional concerns to mention.

1. The authors added a large amount of new results, unfortunately, I did not see all those corresponding results for other related methods, such as turbo results of AutoCoT. Such results are very important given the similarity of these works. In addition, the following trend usually holds: the stronger the LLMs' capability, the less the necessity of labeled data.

2. As LLMs are dominating the field, it is mostly acceptable to challenge a traditional method without being compared with an LLMs-based method. But here I would like to mention the opposite way. Given that hundreds of labeled data are needed, I would be curious what result a supervised method will achieve by using these few hundreds of labeled data for training.

In addition, I have some optional advice. The authors might want to change the name Automate-CoT to something else because it is misleading with Auto-CoT. Given that your method needs hundreds of training data, maybe weakly-supervised-CoT is more appropriate.

---

### Decision · Program_Chairs · 2023-10-07

**Decision:**

Accept-Findings

**Comment:**

This paper presented a work with a few nice contributions as mentioned by the reviewers and summarized by the authors during discussions, which I would prefer not to repeat. I do have a few additional concerns to mention.

1. The authors added a large amount of new results, unfortunately, I did not see all those corresponding results for other related methods, such as turbo results of AutoCoT. Such results are very important given the similarity of these works. In addition, the following trend usually holds: the stronger the LLMs' capability, the less the necessity of labeled data.

2. As LLMs are dominating the field, it is mostly acceptable to challenge a traditional method without being compared with an LLMs-based method. But here I would like to mention the opposite way. Given that hundreds of labeled data are needed, I would be curious what result a supervised method will achieve by using these few hundreds of labeled data for training.

In addition, I have some optional advice. The authors might want to change the name Automate-CoT to something else because it is misleading with Auto-CoT. Given that your method needs hundreds of training data, maybe weakly-supervised-CoT is more appropriate.